# Data-driven approaches linking wastewater and source estimation hazardous waste for environmental management

Wenjun Xie[1], Qingyuan Yu[1], Wen Fang ⬤[1] ✉, Xiaoge Zhang[2], Jinghua Geng[1], Jiayi Tang[1], Wenfei Jing[1], Miaomiao Liu ⬤[1] ✉, Zongwei Ma ⬤[1], Jianxun Yang[1] & Jun Bi ⬤[1] ✉

Industrial enterprises are major sources of contaminants, making their regulation vital for sustainable development. Tracking contaminant generation at the firm-level is challenging due to enterprise heterogeneity and the lack of a universal estimation method. This study addresses the issue by focusing on hazardous waste (HW), which is difficult to monitor automatically. We developed a data-driven methodology to predict HW generation using wastewater big data which is grounded in the availability of this data with widespread application of automatic sensors and the logical assumption that a correlation exists between wastewater and HW generation. We created a generic framework that used representative variables from diverse sectors, exploited a data-balance algorithm to address long-tail data distribution, and incorporated causal discovery to screen features and improve computation efficiency. Our method was tested on 1024 enterprises across 10 sectors in Jiangsu, China, demonstrating high fidelity ($R^2 = 0.87$) in predicting HW generation with 4,260,593 daily wastewater data.

Waste management is increasingly becoming a focal topic in the field of environmental management because it is broadly related to over half of the sustainable development goals (SDGs)[1], such as industry, innovation, and infrastructure (SDG9) and climate actions (SDG13). Hazardous waste (HW) is of priority in waste management due to its harmful properties (i.e., toxicity, corrosiveness, and flammability) as well as its high potential in resource recycling[2]. In the next decades, it was deemed that the world is on a trajectory where waste generation will drastically outpace population growth[3]. The acceleration of industries and the ensuing soaring of HW generation will intensify the burdens of HW management.

Source management of HW, improving the collection rate and reducing generation intensity, is a fundamental process in HW management[4]. However, many countries, particularly those with lax regulations, face challenges in achieving adequate collection,

leading to uncontrolled risk sources and potential environmental damage[5,6]. While in many economies, government agencies mandate hazardous waste generators to pay and transfer their HW to recycle/disposal stakeholders for proper treatment[7–9], the lack of accurate estimation of HW at the firm-level poses a challenge for regulatory stakeholders in optimizing collection systems and inspecting HW generator's behavior. Even though there are some programs, such as the Toxic Release Inventory Program in the United States[10] or the European Pollutant Release and Transfer Register in the EU[11], requiring HW generators to declare their HW generation, the main challenge is to ensure the participation of all generators regularly, particularly when the mandated declaration period is short as week or month. An efficient method to estimate the HW generation which could obtain a detailed and conducive data at the firm-level is an urgent need for HW management.

[1]State Key Laboratory of Pollution Control and Resource Reuse, School of the Environment, Nanjing University, Nanjing, Jiangsu, China. [2]Department of Industrial and Systems Engineering, The Hong Kong Polytechnic University, Kowloon, Hong Kong. ✉e-mail: wenfang@nju.edu.cn; liumm@nju.edu.cn; jbi@nju.edu.cn

Unfortunately, there lacks a low-cost but high-accuracy method to estimate the HW generation quantity at the firm-level across a large region because there are huge disparities in the HW generation patterns among enterprises. The traditional method of estimating the HW generation quantity is based on HW generation intensity factors which are multipliers to predict HW generation from a production unit, such as the number of employees, or economic output[12]. Nevertheless, the HW generation intensity factor obtained by previous studies or the national guideline is usually the aggregated one of an industry, ignoring the heterogeneity across enterprises[13]. In addition, the data of production activities indicated by economic/physical output usually does not have a sufficiently granular temporal resolution to support HW

management practices and sometimes are even inaccessible for environmental stakeholders.

In recent years, the application of the Internet of Things (IoT) sensors in environmental management, particularly the use of IoT sensors for continuous monitoring of wastewater, has provided insight into tackling this intractable issue[14]. The rationale is that there must exist a correlation between wastewater and HW generation data since both of them are directly related to manufacturing processes where the generation of HW and wastewater resulted from the partition of contaminants in liquid and solid phases[15,16]. For example, metal contaminants generated during the electrolysis and pickling process are partly discharged into wastewater and partly into solid phases, forming the HW of anode sludge[17]. We have summarized that almost 43 of

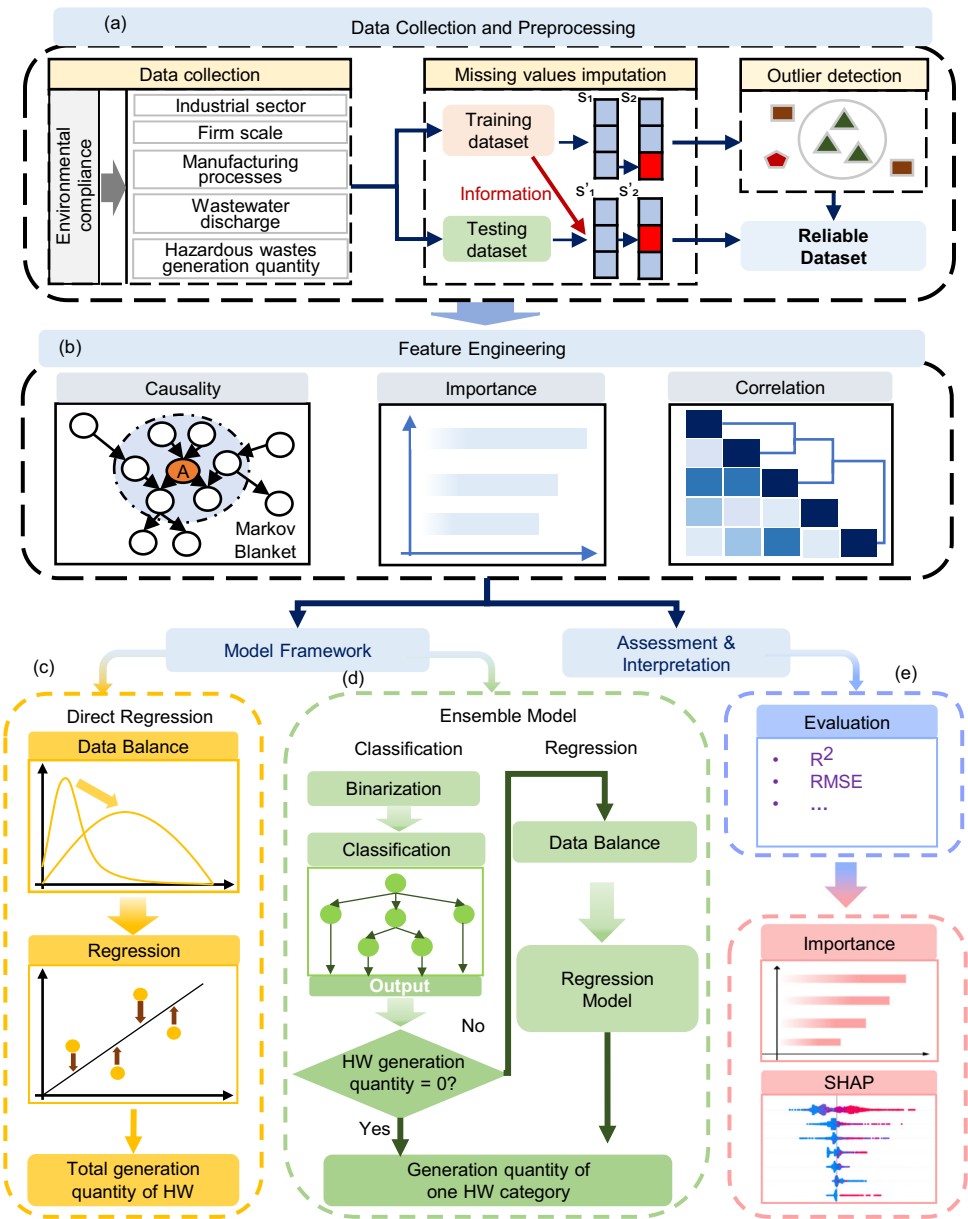

**Fig. 1 | The model framework for predicting the total generation quantity of HW and one specific category. a** Data of variables which could indicate static characteristics and real-time activities of enterprises are collected and pre-processed. **b** Feature engineering which incorporated the causal discovery, importance, and correlation analysis was conducted to screen input features. **c** A regression model is developed directly from the training dataset with data balance to predict the total generation quantity of HW. **d** An ensemble model coupling the classification and regression model is developed to predict the generation quantity

of one category of HW. The binary classification model could determine whether the generation quantity of this category is 0. The regression model is developed to predict the specific value when the generation quantity of this category was >0 based on the classification model results. **e** After model development, performance validation is conducted on the test dataset. In addition, feature exploration is performed on the trained models to investigate the impact of input features on the target output.

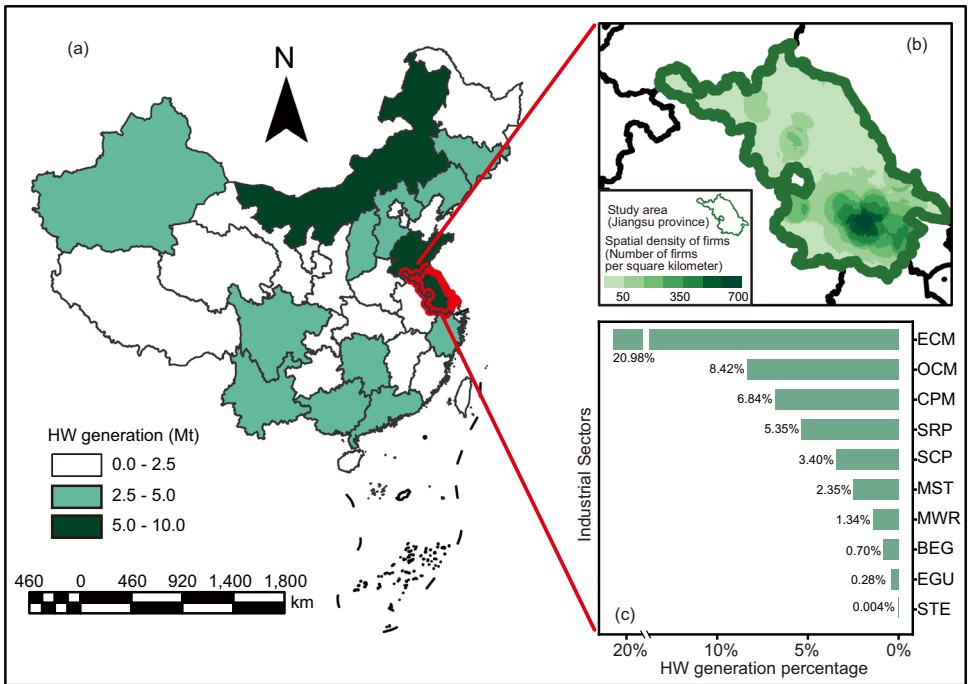

**Fig. 2 | The study region in this study. a** Total generation quantity of China in 2020. **b** The spatial density of 1024 industrial enterprises covered in this study. **c** The top 10 industrial sectors (electronic circuits manufacture (ECM), organic chemical materials manufacture (OCM), chemical pesticides manufacture (CPM), steel rolling and processing (SRP), specialized chemical products manufacture (SCP), metal surface treatment (MST), metal wire and rope manufacture (MWR), biomass energy generation (BEG), electricity generation using other sources (EGU), and steelmaking (STE)) with highest HW generation quantity in the study region.

the total 46 types of HW in the Chinese National List of Hazardous Wastes were generated along with wastewater (Supplementary Table 1). In addition, from the perspective of management practice, compared with economic or physical output data which were commonly adopted to indicate the firm's real-time activities, the data of wastewater was more accessible for environmental management stakeholders and had higher granular time resolution of every hour or day, thereby supporting the source estimation of HW generation with a higher updating frequency.

Here, we proposed a machine learning based and data-driven methodological framework (Fig. 1) with 43 variables to predict the generation quantity of HW at the firm-level every month through linking wastewater data, in tandem with enterprises' static characteristics, to HW generation. To demonstrate the feasibility of this generic framework, an application, using 4,260,593 daily wastewater emission data, was developed to estimate HW generation quantity from 1024 enterprises covering 10 different industrial sectors (Fig. 2). Although the results of our analyses for the 10 industrial sectors were conducted with data from Jiangsu province, China, we discussed how our findings are generalizable to other regions and industrial sectors. This paper offers an effective methodology to fill in the knowledge gap in source management of HW. It also highlighted the potential of data-driven approaches by integrating data cross-organizational boundaries to solve the environmental management challenges in dealing with large-scale heterogeneous stakeholders.

## Results and discussion
### Combined models to predict the generation quantity of hazardous waste at the firm-level
A combined machine learning model was developed to predict the total generation quantity of HW at the firm-level for 1024 enterprises from 10 different industrial sectors across the studied region of Jiangsu, China. Based on the evaluation metric, among these eight machine learning and deep learning algorithms (gradient boosting decision tree (GBDT), support vector machine (SVM), extreme gradient boosting (XGBoost), k-nearest neighbor (kNN), random forest (RF), multilayer perceptron (MLP), MLP ensemble, and tabular neural network (TNN)), the RF model ($R^2 = 0.80$, RMSE = 270.35) achieved the best performance (Supplementary Table 2) and was thus selected for further discussion. Despite employing three deep learning algorithms, their performance in predicting hazardous waste (HW) generation quantity did not surpass that of the RF model. This outcome aligns with findings from a comprehensive study comparing deep learning methods with tree-based models across a standard set of 45 tabular datasets from diverse domains[18]. The study demonstrated that tree-based models, such as RF, remained state-of-the-art for medium-sized data (~10 K samples), even without considering their superior computational efficiency. This superiority can be attributed to specific characteristics of tabular data, including irregular patterns in the target function and the presence of uninformative features. Neural networks are biased to overly smooth solutions, thereby failing to learn non-smooth and irregular data patterns. In contrast, models based on decision trees, which learn piece-wise constant functions, do not exhibit such a bias[19]. In addition, tabular datasets usually contain many uninformative features, but MLP-like architectures are not robust to uninformative features because removing uninformative features will even obviously decrease the model performance[20]. In this study, even for the deep learning model which performed best (TNN: $R^2 = 0.81$, RMSE = 307.85), its performance was comparable to the RF model, but the time cost of model training (TNN: ~5767 s) was much higher than the RF model (12 s). Therefore, considering the RF algorithm's ability to achieve accurate predictions with lower computational costs, we have chosen it for subsequent model development.

Notably, the RF model outperformed the multiple linear regression ($R^2 = 0.22$, RMSE = 592.42) (Supplementary Table 2), underscoring the complex correlation among HW generation, wastewater discharge, and enterprises' static characteristics, which can be effectively simulated by machine learning models. Previous studies on HW generation quantity prediction were commonly on the basis of economic and demographic indicators, such as employee number and product

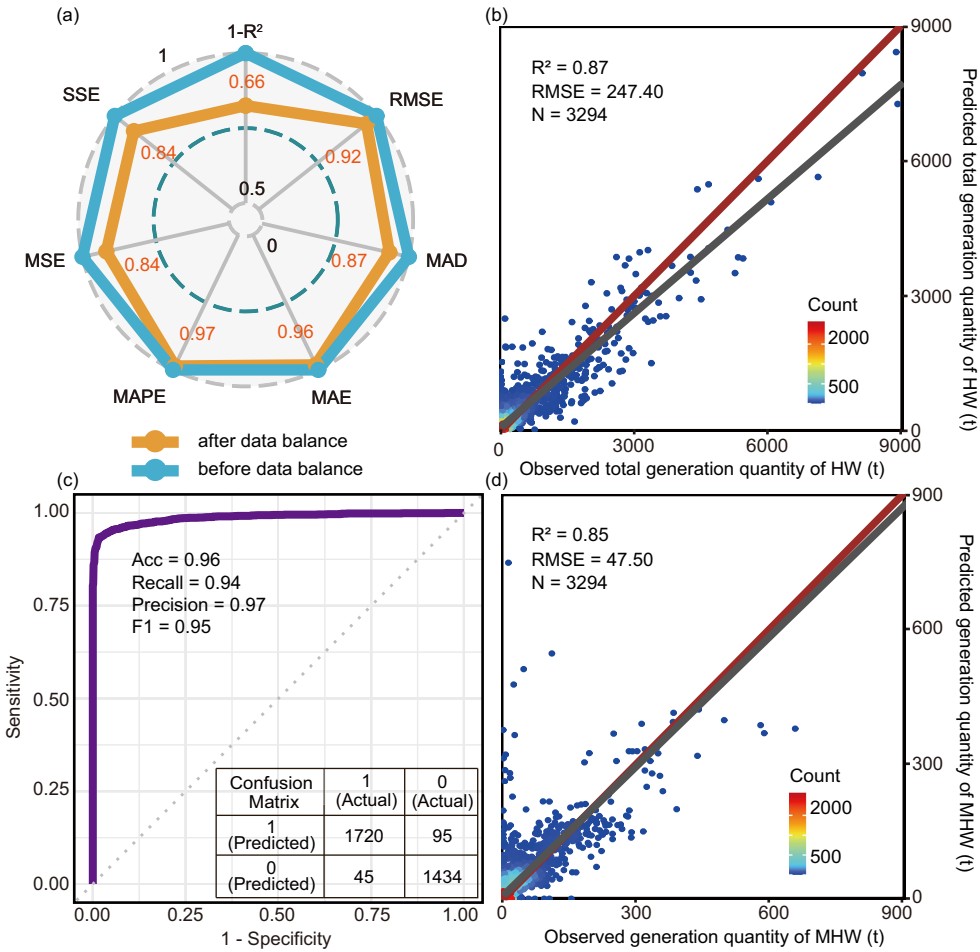

**Fig. 3 | Performance of the combined model to predict total generation quantity of HW and generation quantity of one category (metal surface treatment hazardous waste, MHW). a** Performance comparison of the optimal prediction model using the random forest algorithm for total generation quantity of HW before and after data balancing on the training dataset (using the pre-balancing model as a benchmark and normalizing various indicators). **b** Comparison between the actual total generation quantity of HW and the predicted value by the combined model on the test dataset. **c** The receiver operating characteristic curve of the classification model in the ensemble model for determining whether the value of MHW generation quantity was 0 or >0. **d** Comparison between the actual generation quantity of MHW and the predicted value by the ensemble model on the test dataset.

quantity[12,21]. In this study, we have provided the demonstration of the correlation between HW generation and wastewater discharge.

To further improve the model performance, the data balance method of Synthetic Minority Over-Sampling Technique for Regression with Gaussian Noise (SMOGN) was applied to address the long-tailed distribution of the training dataset where the sample size in the high-value range was disproportionally low (Supplementary Fig. 1). The long-tailed distribution might lower the learning ability of the model in the zone with rare cases. To overcome this limitation, the SMOGN method could generate observations in this zone and increase the fraction of observations with high values in the training dataset (Supplementary Fig. 2). Such a process of data balance is efficient in improving the model performance, with $R^2$ increasing from 0.80 to 0.87 (Supplementary Table 2, Fig. 3a, b). It implied that the synthetic samples generated by SMOGN were reasonable and reliable, offering accurate reflections of the relationship between input features and HW generation. Given that most data in the real practice of environmental management is imperfect, some data processing techniques, such as data re-sampling and synthetic data generation, are prerequisites to the development of a data-driven approach.

Regarding the generation quantity of one HW category, it is different from the total generation quantity because some enterprises might not generate this HW category. That is, the value of generation quantity might be 0 for a considerable fraction of observations. Even for

the enterprise that generated this HW category previously, its behavior will change over time. Aiming at this condition, we took one category of HW, metal surface treatment hazardous waste (MHW, referring to code HW17 in the HW list issued by the Ministry of Ecology and Environment of China)[22], as a case, and an ensemble model coupling the binary classification model and regression model was developed to predict the MHW generation. The binary classification determined whether the value of the MHW generation quantity was 0 and the regression model predicted the specific value when the MHW generation quantity was >0 based on the classification results. During developing the regression model, observations with MHW generation quantity being 0 were removed from the training dataset before data balance.

The RF model demonstrated superior performance in both the binary classification and regression models (Supplementary Table 3), leading to a reliable ensemble model with an $R^2$ of 0.85 and RMSE of 47.50 (Fig. 3d, Supplementary Table 4). Moreover, this ensemble model outperformed a direct regression model (Supplementary Fig. 3, $R^2 = 0.69$, RMSE = 65.80) developed from the whole balanced training dataset containing the observation with MHW generation quantity being 0. The ensemble model's superiority was further explained by comparing their prediction accuracy on the two split test datasets with MHW generation quantity being 0 or >0. For the test dataset with MHW generation quantity being 0, the ensemble presented significantly higher precision than the direct regression model, which

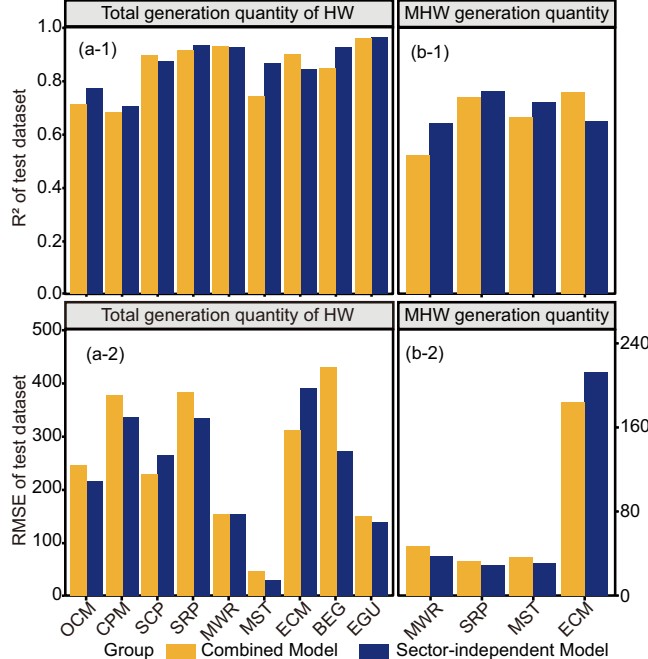

**Fig. 4 | Performance comparison between combined and sector-independent models.** Evaluation metrics of $R^2$ and RMSE on the test dataset for predicting (**a**-1, **a**-2) total generation quantity of HW and (**b**-1, **b**-2) generation quantity of MHW. For the same sector, performances of the two models were evaluated on the same test dataset. Industrial sectors involved organic chemical materials manufacture (OCM), chemical pesticides manufacture (CPM), specialized chemical products manufacture (SCP), steel rolling and processing (SRP), metal wire and rope manufacture (MWR), metal surface treatment (MST), electronic circuits manufacture (ECM), biomass energy generation (BEG), and electricity generation using other sources (EGU).

means that the ensemble model could more accurately screen out the observations with MHW generation quantity being 0 (Supplementary Table 5). For the test dataset with MHW generation quantity >0, the ensemble model still presented significantly better performance (Supplementary Table 5). These optimizations might be because separating the training dataset and developing two models will make each learning goal more focused. There were significant differences between the input data samples with MHW generation quantity being 0 and >0 (Mantel test r = −0.008, p = 0.995, 999 permutations). If these two groups of data were combined as a training dataset, the model would be compelled to learn some redundant information that is not useful for the prediction target. These findings aligned with other studies employing a similar modeling framework coupling classification and regression models[23,24].

## Sector-independent model development
Even though the final combined model can predict the HW generation quantity of enterprises for 10 industrial sectors, its performance may vary significantly among sectors due to diverse patterns in HW generation (Fig. 4(a-1)). For example, the combined model exhibited poor accuracy in predicting the total generation quantity of HW for the sector of chemical pesticide manufacture (CPM, $R^2 = 0.68$). Therefore, independent models for each sector were developed following the same approach.

Considering the good performance of RF compared with other 7 machine learning or deep learning algorithms during the development of the combined model, the RF algorithm was adopted to build independent models for 9 sectors (Supplementary Fig. 4), excluding the sector of steelmaking (STE) owing to insufficient sample size for model development (Supplementary Fig. 1). To ensure a fair comparison

between the predictive performance of the combined model and the sector-independent model, the same test dataset was employed. The comparison showed that performances of independent models were superior to the combined model, particularly for the sector with unsatisfactory prediction accuracy from the combined model (Figs. 4(a-1, a-2), Supplementary Table 6). For example, the $R^2$ of the independent model for the sector of metal surface treatment (MST) ($R^2 = 0.86$) and organic chemical materials manufacturing (OCM) ($R^2 = 0.77$) significantly surpassed that of the combined model ($R^2 = 0.74$ for MST and $R^2 = 0.71$ for OCM, Fig. 4(a-1)). Similar trends can be found for models to predict the generation quantity of MHW. Independent models for MHW generation were developed for 4 sectors (metal surface treatment (MST), steel rolling and processing (SRP), electronic circuits manufacture (ECM), and metal wire and rope manufacture (MWR)) which accounted for 98% of MHW total quantity among the studied industrial sectors (Supplementary Fig. 5). The comparison between the independent and combined models showed that performances of independent models were commonly better than the combined model (Figs. 4(b-1, b-2), Supplementary Table 6). For example, the $R^2$ of the independent model for the sector of metal surface treatment (MST) ($R^2 = 0.72$) was higher than the combined model ($R^2 = 0.66$ for MST, Fig. 4(b-1)). This superiority of independent models was attributed to the high diversity in HW generation patterns among sectors and sector-independent models better capture the unique patterns for each sector. But it is crucial that the data size of each sector must be large enough to provide confidence in learning.

## Influence of input features on predicting hazardous waste generation
The Shapley additive explanation (SHAP) analysis was conducted to interpret the influence of input features on the HW generation, which could provide preliminary insight into the correlation between HW generation and firm's static characteristics and wastewater discharge (Fig. 5).

To clearly illustrate the influence of different features, we classified the predictors into 5 groups: (a) the industrial sector, (b) the firm scale, (c) manufacturing processes, (d) the wastewater routine monitoring indicators (wastewater discharge amount, chemical oxygen demand (COD), pH, ammonia nitrogen ($NH_3$-N), total nitrogen (N), total phosphorus (P)), and (e) metal emission in wastewater (iron (Fe), total chromium (Cr), hexavalent chromium ($Cr^{VI}$), copper (Cu), zinc (Zn), and nickel (Ni)). The average MAS value of all the variables in one group was calculated to quantify the importance of each group. For the combined model to predict the total generation quantity of HW, the importance of these 5 groups were in the following order: firm scale (average relative importance: 16.8%) > the wastewater routine monitoring indicators (average relative importance: 9.5%) >metal emission in wastewater (average relative importance: 8.8%) >sector (average relative importance: 8.7%) >manufacturing processes (average relative importance: 3.6%).

For the firm scale, the samples with low feature values were mainly on the left side, while the points with high feature values were mainly on the right side (Fig. 5b), suggesting the positive relationship between the firm scale and HW generation. It is consistent with the common sense that larger enterprises tend to generate more HW. Some studies adopted the number of employees, the factor used to determine the firm scale in this study, as a production unit to project the waste generation[12].

The wastewater routine monitoring indicators, particularly wastewater discharge amount, held significant importance, emphasizing the close positive relationship between wastewater and HW generation because it was a reflection of contaminant partition between liquid and solid phases. Even so, it is noteworthy that such contaminant partition is highly heterogeneous for different firms (Supplementary Fig. 6) due to diverse manufacturing factors, such as the production technology

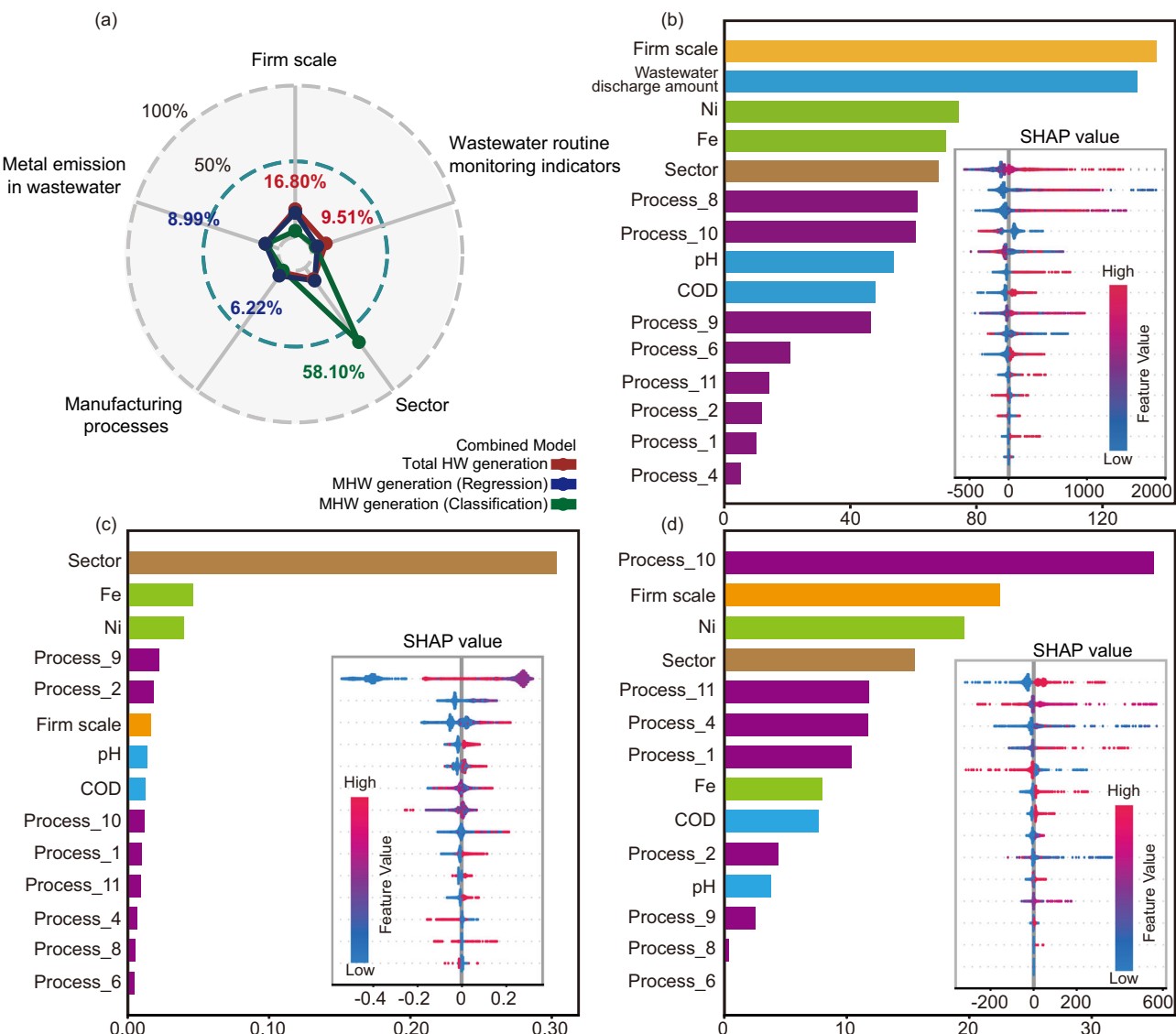

**Fig. 5 | Importance of input features for combined models. a** The average relative importance of variables in 5 groups of the industrial sector, the firm scale, manufacturing processes, the wastewater routine monitoring indicators, and metal emission in wastewater. SHAP summary plots for **b** the combined model to predict total generation quantity of HW, **c** the classification model, and **d** the regression model of the combined ensemble model to predict generation quantity of MHW.

and management level[25]. Even in the same industrial sector, the different techniques and processes adopted during manufacturing can influence the type and concentration of contaminants, and thus their proportional distribution among different phases. For instance, during wastewater treatment, the different techniques of adsorption, coagulation, and electrolysis might alter the form of heavy metal ions in wastewater and the removal efficiency, leading to a change in contaminant partition between wastewater and HW[26]. In the metal processing sectors, different dissolution solvents will impact the extraction efficiency of metals from residues and thereby change their concentrations in wastewater and the quantity of HW[17]. Besides, the environmental management level will impact the compliance of enterprises to the local regulations and accordingly their ability to control pollutant emission[27]. However, in tandem with the data about firm's static characteristics, this heterogeneous relationship can be simulated using machine learning algorithms and used to favor source estimation of HW.

The sector's importance was underscored by the considerable disparities in HW generation quantities across different industrial sectors (Supplementary Table 7). For example, the sector of

steelmaking (STE) had the highest mean value of total generation quantity of HW (995.74-ton average), almost 20 times that of metal surface treatment (MST) (50.64-ton average). The comparable importance observed between variables of metal emission and sector could be attributed to the sector-dependent presence of metals in wastewater. For example, Zn, Fe, and Ni are representative contaminants for the sector of metal surface treatment (MST) due to the electroplating process[28,29], while Cu is discharged in the wastewater for the sector of electronic circuits manufacture (ECM) owing to its extensive use of Cu-containing raw materials[30]. However, the importance of manufacturing processes was generally low compared with other variables.

For the ensemble model to predict the MHW generation quantity, SHAP analysis was conducted for the classification (Fig. 5c) and regression model (Fig. 5d), respectively. For the classification model, the importance of these 5 groups differed from the combined model to predict the total generation quantity of HW and ranked as follows: sector (average relative importance: 58.1%) >metal emission in wastewater (average relative importance: 8.2%) >firm scale (average relative importance: 3.2%) >the wastewater routine monitoring indicators

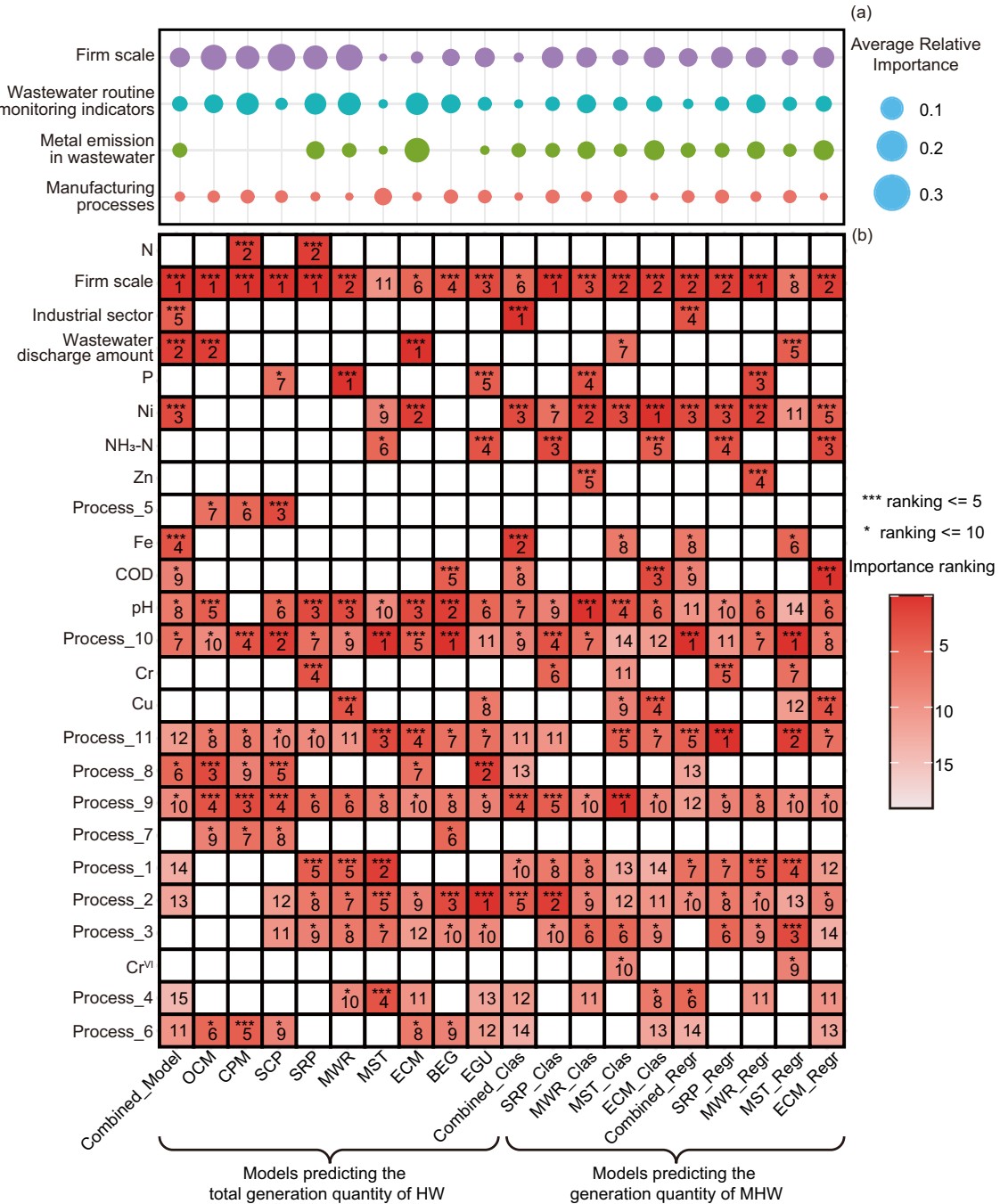

**Fig. 6 | Importance of input features for 20 models, including a combined model and 9 sector-independent models to predict total generation quantity of HW, 5 classification models and 5 regression models from the combined and independent ensemble model to predict MHW generation quantity. a** The average relative importance, indicated by the average MAS value of variables in 4 groups of the firm scale, manufacturing processes, the wastewater routine monitoring indicators, and metal emission in wastewater. **b** Ranking of input feature importance for 20 models. The number inside each cell reflected the ranking of the variable. Variables ranked in the top 5 are marked with *** to denote their crucial importance in model prediction, while variables ranked between 6th and 10th are marked with * to indicate their relatively significant impact.

(average relative importance: 2.6%) >manufacturing processes (average relative importance: 2.2%). In this classification model, the role of sector and metal emission in wastewater became considerably more crucial in determining whether the firm will generate MHW. The sector is the most dominant factor since the categories of HW generated in one sector are almost identical. In this study, most of MHW were generated in the four sectors of metal surface treatment (MST), steel rolling and processing (SRP), electronic circuits manufacturing (ECM), and metal wire and rope manufacturing (MWR).

Regarding the regression model in the ensemble model to predict MHW generation quantity, the importance of these 5 variable groups ranked as firm scale (average relative importance: 14.6%) >sector (average relative importance: 10.1%) >metal emission in wastewater (average relative importance: 9.0%) >manufacturing processes (average relative importance: 6.2%) >wastewater routine monitoring indicators (average relative importance: 3.8%). In comparing the two regression models for predicting total HW generation quantity and MHW generation quantity, a notable distinction emerged in the

relative importance of metal emission in wastewater and wastewater routine monitoring indicators. Specifically, compared with wastewater routine monitoring indicators of COD, the higher importance assigned to metal emission in wastewater in predicting MHW generation aligns with the established understanding that MHW, generated during metal surface treatment, exhibits strong correlations with metals present in wastewater. This underscores the significance of considering metal discharge levels in predicting MHW generation, as compared to other common contaminants. Although the average importance of manufacturing processes appeared relatively low, a specific variable of the manufacturing process of wastewater treatment (process_10) stood out. This variable played a notably high role, which can be attributed to the fact that a significant portion of MHW is derived from the treatment of metal wastewater[22,31].

To further uncover the generalized relationship between HW generation and input features of firms' static characteristics and wastewater discharge, we compared the feature importance among 20 models, including a combined model and 9 sector-independent models to predict total generation quantity of HW, 5 classification models and 5 regression models from the combined and independent ensemble model to predict MWH generation quantity (Fig. 6). Since the variable of the sector was not involved for the independent models, 4 groups of (a) the firm scale, (b) manufacturing processes, (c) the wastewater routine monitoring indicators, and (d) metal emission in wastewater were discussed.

The comprehensive analysis across 20 models highlighted the consistent importance of firm scale and wastewater routine monitoring indicators, including wastewater discharge amount and emission of common pollutants (i.e., N, P, COD, and NH$_3$-N). Nonetheless, the importance of manufacturing processes and metal emission in wastewater was sector-dependent. For instance, Zn exerted greater influences on predicting HW generation in the sector of metal wire and rope manufacture (MWR), but Cu was more important for the electronic circuits manufacturing (ECM) sector. For some sectors, the variable group of metal emission have been even filtered out during feature screening incorporating causal discovery and feature importance. Such heterogeneity was also significant for manufacturing processes. For example, for the sector of metal surface treatment (MST), machining processing (process_1) held significant importance as it leads to the generation of various HW, including waste acids (referring to code HW34), waste alkali (referring to code HW35), metal surface treatment waste (referring to code HW17), and hazardous lubricants (referring to code HW08)[32,33]. In addition, the rectification and distillation process (process_5) was pivotal to HW generation in the specialty chemical products (SCP) sector because the spent chemical solvent (referring to code HW06) generated during this process was one representative type of HW in the chemical industries[34].

## Generalizability of this model framework

To demonstrate the adaptability of our generic model framework in predicting HW generation across diverse regions and industrial sectors, we developed 3 case study applications (Supplementary Fig. 9). The first two cases were crafted to illustrate the model's applicability in regions with varying contaminant partition between liquid and solid phases influenced by regional water withdrawal strategy and technology advancement (Supplementary Fig. 10). The third case was designed to show the feasibility of the generic framework in more representative sectors.

Specifically, the first application focused on the development of a model for the sector of metal surface treatment (MST) in Shandong province which had a lower ratio of wastewater discharge amount to HW generation quantity (WTH) than the studied region of Jiangsu. The ratio of WTH could indicate different contaminant partition relationships between the solid and liquid phases. The sector of MST was chosen because it had sufficient sample sizes to build models for most provinces. The second focuses on model development in Zhejiang province which had higher WTH than the studied region of Jiangsu. The third focuses on another industrial sector of lead and zinc metallurgy sector in Hunan province, not studied in Jiangsu but significant for HW generation in China. Due to the data availability, these three cases utilized the HW generation and wastewater data with time resolution of the year from the China Environmental Statistics Database of 2015. Detailed data information about the three cases can be found in Supplementary Text 1.

For the first two cases developed for regions with different WTH ratios, models presented commendable predictive performances (Shandong province: R$^2$ = 0.69, RMSE = 29.38; Zhejiang province: R$^2$ = 0.72, RMSE = 721.15, Table 1). This suggested that our model framework adeptly captured and simulated the heterogeneous relationship between wastewater and HW generation, even when it varied across regions. The relatively lower predictive performances observed in these applications, compared with the previous model (R$^2$ = 0.87, Supplementary Table 2), developed from monthly observations, may be primarily attributable to the smaller data size available for these cases (the Shandong case: 190 observations; the Zhejiang case: 396 observations). It is important to acknowledge that collecting more data has the potential to enhance the performance of data-driven approaches. The expansion of the dataset can compensate for the loose structure of the current smaller dataset and address the omission of potential information. Consequently, this can improve the representativeness of the data, enabling the underlying model to better characterize the actual distribution[35]. Afterwards, in the Zhejiang case, the model developed for this region was applied to predict HW generation in Shanghai, a region in the same cluster as Zhejiang based on the WTH ratio. However, the prediction performance was below expectations (R$^2$ = 0.32). This implied that the model must be trained using the localized data derived from the application region. Concerning the third case focusing on the sector of lead and zinc metallurgy, the model still exhibited high reliability and prediction accuracy with the R$^2$ and RMSE values of 0.82 and 2366.63, respectively (Table 1). This reaffirmed the potential of our framework for application in diverse regions and industrial sectors, provided that model training data is localized, and variables are appropriately screened based on the specific studied sector.

Besides, since it has been demonstrated that the feature importance varied with sectors, we further investigated whether their

**Table 1 | Model performances of three application cases in other regions or sectors**

| Region | Sector | R$^2$ | RMSE | Final input features |
|---|---|---|---|---|
| Zhejiang, China | Metal surface treatment | 0.72 | 721.15 | manufacturing processes (including: metal surface processing, circuit board treatment, equipment maintenance, exhaust gas treatment, and wastewater treatment), firm scale, wastewater discharge amount, N, P, NH$_3$-N |
| Shandong, China | Metal surface treatment | 0.69 | 29.38 | manufacturing processes (including: metal heat treatment, metal surface processing, circuit board treatment, equipment maintenance, exhaust gas treatment, and wastewater treatment), firm scale, COD, NH$_3$-N, N, Cr$^{VI}$, Cr |
| Hunan, China | Lead and zinc metallurgy | 0.82 | 2366.63 | manufacturing processes (including: mineral leaching, electrolysis, roasting, and flue gas treatment), firm scale, COD, NH$_3$-N, P, Cr$^{VI}$ |

importance will change with regions even within the same sector, thereby providing guideline for the variable screening when the generic model was applied in different regions. To have a better understanding of the spatial heterogeneity, due to data availability, we expanded the model of the MST sector to include three additional regions (Guangdong, Hebei, and Fujian province) using the data from the same database of the China Environmental Statistics Database of 2015. Detailed information about the model performance for these cases can be found in Supplementary Table 9. Finally, we compared the feature importance of six models developed for the MST sector across the regions of Shandong, Zhejiang, Jiangsu, Guangdong, Hebei, and Fujian provinces. Upon comparison among the six regions, we observed consistently high importance attributed to wastewater routine monitoring indicators (wastewater discharge amount, COD, NH$_3$-N, P, and N), with the average relative importance of variables in this group ranking 1st for most regions (Supplementary Fig. 12). However, significant disparities in feature importance were noted for the groups of manufacturing processes and metal emission in wastewater. For example, while the variable group of metal emission in wastewater (Cr$^{VI}$ and Cr) was most important for the Hebei province, this group of variables were filtered out during feature screening based on causal discovery and feature's importance for the Zhejiang province. This discrepancy was because the enterprises of the MST sector in Hebei province are mainly related to chrome plate, a subsidiary sector of the leather industry which has been listed as one of the pillar industries in this region[36]. Another finding is that a distinctively high importance of manufacturing processes was observed for Guangdong province, owing to its unique industrial layout where electronic waste recycling is well developed. Enterprises engaged in the manufacturing process of circuit board treatment contribute significantly to HW generation in this region[37]. The heterogeneity in feature importance across different regions, even for the same sector, underscores the importance of employing a generic set of variables tailored to the characteristics of sectors in different regions. Subsequently, specific variables used to build models can be screened based on feature engineering, considering the varied importance of features for each application region.

## Sensitivity and uncertainty analysis

We employed a series of sensitivity analyses to assess the robustness of our model. First, feature selection plays a crucial role in model development, and an inappropriate screening of features may adversely affect the model's performance. We evaluated the effect of different feature selection principles, including (i) screening features based on Markov Blanket (MB) according to the directed acyclic graph (DAG) learning and (ii) combining the features screened based on MB and features' importance ranking, on the model performance. The comparison showed that relying solely on MB-based features sometimes led to a decrease in predictive performance (Supplementary Tables 10 and 11). For example, in the combined model to predict the total generation quantity of HW, using the MB-based features diminished the R$^2$ from 0.87 to 0.68 on the testing dataset. This reduction was because the MB learning algorithm might fail to identify some critical features during feature selection tasks caused by the strict assumption of data distribution, variable types, or correctness of criteria, during causal discovery, which thereby limited the faithfulness of DAG learning results[38]. However, when the selected features were further refined based on the features' importance, the model even could achieve superior performance to the baseline models with all variables as input. For instance, the R$^2$ of the independent models for the sector of biomass energy generation (BEG) (R$^2$ = 0.92) surpassed the baseline models with all variables as input (R$^2$ = 0.87). In addition, it is worth highlighting that the feature selection significantly increased the computational efficiency by decreasing the total computation time of training combined and sector-independent models by 20% (from 158.1s to 125.7 s). Therefore, considering the computation efficiency as

well as the model performance, it is satisfactory to select features through combining MB learning and features' importance analysis.

Second, we estimated the temporal extrapolation of the model during the application, referring to how the performance of the model changed with prediction times because the model was commonly built using the historic data but used to predict the current HW generation in management practice. The change in technique advancement and manufacturing activity level will impact the predictive performance of the model built on historical data. Here, the model, trained using data from Jan. 2020 to Dec. 2021, was employed to predict HW generation for each month in 2022. As shown in Supplementary Fig. 14, the combined model consistently performed well with an R$^2$ around 0.7 when it was used to predict the total generation quantity of HW in the next 3 months. Then, with the further extrapolation of time, the performance gradually decreased to R$^2$ of 0.4–0.5. This might be attributable to a shift in the HW generation pattern not learned by the model using historical data. A similar trend can be found for the combined model to predict the generation quantity of MHW. Therefore, we recommended periodic retrofitting and retraining of the model every three months to ensure accuracy during application.

To further ensure the reliability of the model in real-world applications, it is essential to estimate model uncertainty because in practice, decision-making in the context of industrial applications involves a complicated trade-off between risky decisions and large potential economic benefits. ML models have undoubtedly advanced the field by offering a valuable set of tools to efficiently learn from data and automate the management process. Nevertheless, these models require effective uncertainty quantification to demonstrate their trustworthiness. Ideally, the uncertainty quantification of these ML models should yield a c% confidence interval that contains the true value for approximately c% of the time[39]. For example, if c% = 95%, we expect that approximately 95% of test samples have their true values fall into the respective 95% confidence intervals of prediction. Our uncertainty analysis indicated low uncertainty in model predictions during testing. For the combined models to predict HW generation quantity, 95.60% of testing samples fell into the 95% confidence intervals of prediction (Supplementary Table 12, Supplementary Fig. 15). This implied the high reliability in the model prediction results and good generalization of models. However, regarding the sector-independent models and the model to predict MHW generation quantity, some of them exhibited overconfidence, with observed confidence levels lower than the expected 95%. It might be attributable to the small size and long-tail distribution of data for learning, and obtaining more reliable data could decrease uncertainty.

## Cost-effectiveness analysis

The cost-effectiveness of this scheme was compared with traditional management measurement of field surveys from the perspective of time and monetary cost. The time associated with developing the model was evaluated. The framework was implemented on a personal computer of configuration Intel Core i5-1135G7 with CPU 2.4 GHz and 16 GB memory. The model development time was affected most by the matching and integration of enterprise wastewater data and HW generation data, and it took a total of 13 min 22.7 s to complete the program for the 10 industrial sectors with 4,260,593 data points. In contrast, the data preprocessing of missing value imputation and outlier detection for the same 10 industrial sectors, could be accomplished in 35.1 s. Feature engineering took a total of 171.2 s. The subsequent machine learning model construction program, including data balancing, model training, and feature inference, consumed a total of 125.7 s. Generally, the model development will cost about 20 min and the retrofit of the model will take 12 min every 3 months according to the data size. However,

traditional field surveys to determine each enterprise's HW generation intensity factor, crucial multipliers for predicting HW generation from a production unit, were time-consuming, often requiring more than one day for a single enterprise[40]. The time will be even longer when the investigator is not professional with the manufacturing processes of HW generation.

Regarding the expenditure, this scheme will not augment the company's expenditures because the data used to build the model was the wastewater data monitored by IoT sensors which have been mandated to be installed in many countries. For example, Vietnam's Environmental Protection Law requires enterprises to install automatic pollution monitoring equipment[41]. Even in cases where automatic monitoring sensors are not mandated to be installed, regulatory authorities commonly require industrial enterprises to periodically monitor and report wastewater data[42,43]. Utilizing this data, it is still feasible to predict the HW generation during the same period as the reported time of wastewater data. Meanwhile, with the regulation being increasingly stringent, mandating enterprises to report near-real-time information about wastewater discharge and installing automatic sensors to monitor wastewater will become unavoidable[44]. Even so, it has been demonstrated that adopting these sensors can result in cost savings because the real-time data combined with intelligent algorithms will favor the optimization of system operation to save energy costs[45].

### Model applicability, limitation, and future research

Our generic framework has been demonstrated being applicable in diverse regions and industrial sectors, provided that model training data is localized, and variables related to manufacturing processes and water contaminant emission are appropriately screened based on the specific studied sector. For one industrial sector, a generic set of variables tailored to the characteristics of sectors should be employed for different application regions and then specific variables used in each region can be screened based on feature engineering, considering the varied importance of features. In addition, since the model simulates the patterns of HW generation at the firm-level through mining data characteristics, it is suggested to collect as much data as possible to enhance model performance and reliability.

When the model was implemented in the real management practice, firstly, the variables involved in the model, referring to the ones related to enterprise characteristics (such as the industrial sector, firm scale, and manufacturing processes) and real-time water contaminant emission, should be tailored based on the characteristics of studied sectors. Regarding the tailored variables, a set of historic data, with a monthly time-resolution, should be collected for subsequent modeling development. Secondly, apply a data-driven methodology incorporating feature engineering, data balancing, and artificial intelligent algorithms to build the model for predicting HW generation quantity at the firm-level for each month. Given the high diversity in HW generation patterns among sectors, it is advisable to develop sector-independent models that can better capture the unique patterns of each sector when the data size for each sector is sufficiently large to ensure confidence in learning. Thirdly, adopt this model to predict the enterprise's HW generation quantity in the next 3 months following the period of collected historic data. In addition, regular updates of the model every three months are recommended to maintain model performance.

Some limitations remain and deserve further study. Firstly, the case in this study was a regional survey involving enterprises from only 10 sectors. While a comprehensive analysis of hazardous waste (HW) generation characteristics across representative sectors has been undertaken, further applications in diverse regions and sectors are needed to adapt the model. Secondly, considering the data availability and the model's applicability to environmental regulatory, some crucial variables, such as raw material inputs, product outputs, and

electricity consumption, were not involved in the framework because acquiring high-resolution data for these features remains challenging. However, with the advancement in the realization of big data across various domains, future iterations of the model framework can incorporate additional variables to improve its accuracy.

## Methods

The model development process comprised three main stages of data collection and processing, feature engineering, and model construction (Fig. 2). Firstly, the original whole dataset was randomly split into a training dataset and a testing dataset before performing any preprocessing steps to avoid data leakage[46,47]. Then preprocessing steps, including missing value imputation and outlier rejection at the ratio of 5% using unsupervised machine learning algorithms, were applied to the training dataset. This resulted in an 8:2 ratio for the sizes of the training dataset and the testing dataset. Notably, missing values in both the training and testing dataset were imputed based on the information from the training dataset to further control the risk of information leakage[48]. Secondly, feature engineering, incorporating causal discovery, importance ranking, and correlation analysis, was implemented to select a subset of informative features for building following models. Thirdly, the training dataset was balanced to modify the long-tailed distribution and the machine learning/deep learning models were trained and evaluated based on their performance on the testing dataset.

### Variables and data

Our study aims to build a generic data-driven model framework to simulate the patterns of HW generation at the firm-level at a large scale. Previous studies have demonstrated that HW generation shows strong affinities with both static enterprise characteristics (such as sector, manufacturing processes, firm scale) and real-time manufacturing activities[49,50], so 43 variables (19 variables related to enterprise characteristics and 24 variables related to real-time manufacturing activities) were utilized to predict the HW generation in this framework. For specific cases, features could be screened from these 43 variables and tailored based on the characteristics of studied sectors and data availability.

These 19 variables related to enterprise static characteristics encompassed the industrial sector that the firm belongs to, the firm scale, and 17 manufacturing processes. We employed the number of staff, which has been widely used as an indicator of manufacturing activities, to determine the firm scale and it was classified into five categories (Supplementary Table 13) according to Chinese Criteria for the Division of Large, Medium and Micro Enterprises[51]. Features of 17 manufacturing processes (Supplementary Table 14) were binary variables which were determined as 1 if the firm has this manufacturing process; otherwise, the values were 0. These processes, summarized according to the Chinese Environmental Management Guideline for Hazardous Wastes[52] and literature review, were the dominant manufacturing processes that could generate HW for 50 industrial sectors with the highest HW generation quantity in China (Supplementary Table 15). These 50 sectors, classified based on the 4-digit National Standard Industrial Classification, contributed to 94.2% of the total HW generation quantity in China in 2015 according to the China Environmental Statistics Database. These industrial sectors mainly involved mining (e.g., oil and gas extraction, non-ferrous and non-metallic mineral extraction), industrial manufacturing (e.g., paper and paper production, oil and coal processing, chemical and pharmaceutical manufacturing, metal smelting and processing, and electronic equipment manufacturing), and electricity/heat production.

24 variables used to indicate the firm's real-time activities were features of water contaminant emission, consisting of wastewater discharge amount, COD, pH, $NH_3$-N, N, P, suspended substance, petroleum hydrocarbon pollutants, biochemical oxygen demand, volatile phenol, total organic carbon, sulfate, fluoride, cyanide, lead, arsenic,

cadmium, mercury, Fe, Cr, Cr$^{VI}$, Cu, Zn, and Ni). Wastewater was also the by-product of manufacturing activities and has been demonstrated to have a relationship with HW generation[53,54]. These 24 indicators were mandated to be monitored automatically or manually for various sectors based on national or industrial regulations (Supplementary Table 15). They showed tight affinities with most of the HW categories according to engineering and chemical experiences (Supplementary Table 1). For instance, the textile industry employs synthetic dyes and chemicals during manufacturing, containing heavy metals (e.g., Cu and Zn) and complex organic compounds[55], which will impact the contaminants (e.g., COD, NH$_3$-N, and heavy metal) in wastewater as well as the emerge in HW (e.g., adsorbents and sludge). Considering that these 43 variables were developed based on the HW categories and dominant generation sectors in China, we compared the list of HW issued in China with that of the United States and EU. It could be found that most of the HW issued by the United States and EU were involved in the Chinese list, thereby suggesting the potential of this framework to be applied to address the global issue (Supplementary Table 16).

Based on this generic framework, we took 1024 enterprises from 10 industrial sectors with the highest HW generation quantity in Jiangsu Province, China as cases to develop the model (Fig. 2). China, as the second largest producer of toxic substances after the United States[56], contributed 2.1% of global hazardous waste generation (9.52 million tons) in 2016[57]. The study region of Jiangsu is representative since it is one of the most industrialized provinces in China and had a HW quantity mass of ~5,220,500 tons in 2020, ranking 3rd in China. Such a high generation quantity necessitates the urgent need to strengthen the source management of HW[58]. These selected 10 industrial sectors (Supplementary Table 17) contributed to 49.7% of the total generation quantity of HW in the studied region in 2020. For this case, 25 variables, including the industrial sector that the firm belongs to, the firm scale, 11 manufacturing processes, and 12 variables of wastewater monitoring indicators (wastewater discharge amount, COD, pH, NH$_3$-N, N, P, Fe, Cr, Cr$^{VI}$, Cu, Zn, and Ni), were screened from the total 43 variables in the framework based on characteristics of involved sectors and data availability. The data from Jan. 2020 to Dec. 2022 was collected. The data of employee numbers and industrial sectors were officially from the Department of Ecology and Environment of Jiangsu Province. The data of manufacturing processes for each enterprise were determined by combining the typical processes related to HW generation (Supplementary Table 14) and the information declared by firms. The HW generation data with time-resolution of months was declared by the enterprise, excluding those with records of environmental violations and penalties. 4,260,593 daily wastewater emission data, monitored by automatic IoT sensors, were aggregated monthly and merged with the HW generation data to create monthly observations for each enterprise. Detail information about data sources, quality control, and processing can be found in the Supplementary Text 2 to 5.

After data processing, a total of 16,477 observations were obtained. Sample sizes across different industrial sectors, firm scale, and manufacturing processes are detailed in Supplementary Fig. 17. Descriptive statistics of numeric variables were summarized in Supplementary Tables 7, 8 and 18. In this study, the response variables were the total generation quantity of HW and the quantity of MHW, the most common category with 53.7% of observations generating this category of HW.

## Feature engineering
During artificial intelligent model development, using all available features as input may be prohibitively expensive, unnecessarily wasteful, and may lead to poor generalization performance, especially in the presence of irrelevant or redundant features. Thus, selecting a subset of informative features for building artificial intelligent models has become a standard preprocessing step. In this study, to simplify

the artificial intelligent model and improve its performance, feature engineering that incorporated causal discovery, importance ranking, and correlation analysis was conducted to screen useful input features. Firstly, the causal relationship was analyzed via DAG learning, where we derived a set of cause-and-effect relationships from the observational data, and the minimal set of features with high relevance to the response variable was obtained based on the MB search algorithm. Then, features with high importance in model prediction but not identified as relevant features were added.

For causal discovery purposes, Bayesian networks have been pervasively adopted in the literature because they provide a compact and sound theoretical framework for modeling causal relationships and reasoning with uncertainty over a set of random variables through a DAG[59]. Under certain conditions, the edges in a Bayesian network have causal semantics, thus enabling cause-and-effect analysis to some degree. Bayesian networks, in the form of DAGs, are learned using the constraint-based methods established upon conditional independence testing and search-and-score methods[60]. Due to the combinatorial nature, these methods are computationally demanding as the number of DAGs grows exponentially with the number of observed variables. In this study, we adopted a DAG learning approach converting the traditional combinatorial optimization into a continuous constrained optimization problem to avoid the combinatorial nature and circumvent the enormous computationally demand[61]. Based on the discovered causal DAG, the set of features with high relevance to the response variable was obtained based on the MB search algorithm. An MB was defined as the union of the parents (nodes connected above), children (nodes connected below), and other parents of those children[62] (Supplementary Figs. 18–21).

Features selected based on MB were further adjusted based on the features' importance and correlation. The adjustment focused on the features with high importance in model prediction but not identified as relevant features based on DAG learning and MB selection. Feature importance was analyzed using the SHAP method[63] which provides a way to estimate the contribution of each feature from the perspective of game theory. In the SHAP analysis, the contribution of each feature to the model output is assigned according to its marginal contribution. For each data point, a SHAP value is calculated to describe its impact on the model output[64]. All SHAP absolute values of the points were averaged as MAS values to describe the overall impact of the feature on the model output. The feature with a higher MAS is considered to have a more significant impact on the model output. Variables whose importance ranked top 10 among all the input features were regarded as the ones with high importance.

For these important variables that were not identified during the MB search algorithm, on the one hand, the ones demonstrating a low correlation (Spearman's correlation <0.6) with the features screened using the MB search algorithm could be added to the set of input variables. On the other hand, the variables that had a high correlation (Spearman's correlation >0.6) with the features screened using Markov blanket algorithm as well as a higher importance ranking could be used to replace the correlated ones. To simplify the feature screening processes, only one feature was added or removed at a time and the iteration stopped until the performance did not change significantly.

## Development of the combined model to predict the total generation quantity of hazardous waste based on data balance
After feature engineering, artificial intelligent models were constructed after data balance was performed on the training dataset to modify the long-tailed distribution using the SMOGN method. SMOGN is a data balancing technique that combines under-sampling with two over-sampling techniques of Synthetic Minority Over-Sampling Technique for Regression (SMOTER) and introduction of Gaussian noise[65]. SMOTER generates new synthetic examples from rare cases through an interpolation strategy. This interpolation is carried out using two

rare cases (one is a seed case and the other is randomly selected from the k-nearest neighbors of the seed) and the new synthetic sample is determined as a weighted average of the target variable values of the two rare cases. All rare cases are used in turn as seed examples. Furthermore, the key idea of the SMOGN algorithm is to limit the risks posed by SMOTER through using the more conservative strategy of introducing Gaussian noise. Through introducing Gaussian Noise, new synthetic examples are generated using SMOTER only when the two rare cases selected are 'close enough' and will use the introduction of Gaussian Noise when the two examples are 'more distant'.

To be specific, SMOGN divided the dataset into the rare and normal partitions using a threshold value which was defined as the three-quarters quantile of the response variable in this study. The rare partition was the zone with rare samples being far from the center of the parts. Random under-sampling was applied to the normal partition. The over-sampling procedure of SMOTER interpolation or the introduction of Gaussian noise was applied to the rare partition based on the distance between the seed example and k-nearest neighbors[66]. SMOTER interpolation was used if the distance was less than half of the median distance between the seed example and other data points in the rare partition. Otherwise, a new sample was generated by introducing Gaussian noise on the seed case.

Subsequently, 8 machine learning and deep learning algorithms were used to develop models based on the balanced training dataset (Supplementary text 6). 10-fold cross-validation was applied to the training dataset and the optimum hyperparameters were the ones that achieved the best validation performance (average performance on the 10 validation sets) for regression models which was evaluated using multiple metrics of the coefficient of determination ($R^2$), root-mean-square-error (RMSE), mean absolute error (MAE), mean absolute percentage error (MAPE), median absolute deviation (MAD), mean square error (MSE), and sum of squares due to error (SSE)[67]. The classification model adopted accuracy, recall, precision, and F1-score (F1) as evaluation metrics[14] (Supplementary text 7). Then the optimum model was screened based on their performance on the testing dataset.

### Development of the combined model for generation quantity of one category of hazardous waste

An ensemble model coupling the classification and regression model (Fig. 2d) was developed to predict the generation quantity of one category of HW, referring to MHW in this study. The binary classification model could determine whether the value of MHW generation quantity was 0. The data used to train the binary classification model was the whole training dataset before the data balance. The regression model was developed to predict the specific value when the MHW generation quantity was >0 based on the classification model results. The data used to train the regression model was the observations which had MHW generation quantity >0 among the training dataset. Similarly, the screened observations were also balanced using the SMOGN method for regression model development.

The binary classification and regression model were combined for performance testing. Specifically, the test dataset is first predicted by the classification model. If the observation was predicted to have the value of MHW generation quantity >0, then the regression prediction model will be conducted to predict the specific quantity. Otherwise, the value of the MHW generation quantity will be 0. Finally, multiple metrics ($R^2$, RMSE, MAE, MAPE, MAD, MSE, and SSE) were determined for the testing dataset to evaluate the performance. This ensemble model framework can also be feasible in predicting the generation of other categories of HW.

### Development of the independent model for each industrial sector

During the independent model development, the feature of the sector that the firm belongs to was eliminated from the model input so that there were 30 predictors in total. Independent models to predict the total generation quantity of HW were built for 9 sectors, excluding the sector of steelmaking (STE) with only 20 observations. With regard to the MHW generation quantity, the independent prediction model was developed for only 4 sectors (metal surface treatment (MST), steel rolling and processing (SRP), electronic circuits manufacture (ECM), and metal wire and rope manufacture (MWR)) which accounted for 98% of MHW quantity in 10 sectors. The development of independent models followed the same approach as the combined models.

### Uncertainty analysis

The study utilized Quantile Regression Forest (QRF) to estimate the model's uncertainty in calculating the 95% confidence intervals of the prediction value[68]. Commonly, the RF model calculates its final prediction by weighting the average of each leaf of each tree. However, when estimating the uncertainty, the QRF retained the value of all observations on each leaf, instead of just their mean. Therefore, the QRF estimated the conditional distribution function of the model outputs based on all observations retained in each leaf. Then, a 95% confidence interval was constructed based on this conditional distribution function to estimate the uncertainty in the model output[68].

## Data availability

Subject to privacy and licensing agreements, data can be obtained by request to the corresponding authors.

## Code availability

All code necessary to reproduce the analysis is made available on Github (https://github.com/Monchiwjxie/Hazardous-waste-generation.git) and Zenodo (https://doi.org/10.5281/zenodo.11487629)[69]. Data analysis was performed using Python version 3.9.2 and R version 4.3.0.

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

## Acknowledgements

This work was funded by the National Natural Science Foundation of China (grant No. 52270199; W.F.), National Natural Science Foundation of China (grant Nos. 71921003 and 72234003; J.B.), National Natural Science Foundation of China (grant Nos. 72222012 and 72174084; M.L.), Jiangsu R&D Special Fund for Carbon Peaking and Carbon Neutrality (No. BK20220014; J.B.), and a grant from the Research Grants Council of the Hong Kong Special Administrative Region, China (No. PolyU 25206422; X.Z.).

## Author contributions

W.X., Q.Y., W.F., M.L., and J.B. designed research; W.X., Q.Y., J.G., X.Z., J.T., and W.J. performed research; W.X., Q.Y., X.Z., W.F., J.G., J.T., and W.J. analyzed data; W.F., X.Z., M.L., Z.M., J.Y., and J.B. edited the paper; and W.X. and W.F. wrote the paper.

## Competing interests

The authors declare no competing interests.
