## [Peer Review File · Nature Communications]

Data-driven approaches to bestow environmental management through linking wastewater data to source estimation of hazardous wasteEditorial Note: Parts of this Peer Review File have been redacted as indicated to remove third-party material where no permission to publish could be obtained.

REVIEWER COMMENTS

Reviewer #1 (Remarks to the Author):

The study is done thoroughly and can have relevance for industrial stakeholders and other policy makers. I have few doubts about this MS, please revise as per the given comments/suggestion.

In this study, investigations on a data-driven model are reported for the prediction of hazardous wastes for a large dataset of industries in China.

- The introduction section is too lengthy, readers may lose interest while going through this part. Please condense it remove the redundant information.
- The number of variables considered in this study for enterprise characteristics and real time manufacturing process may not be sufficient for model validation. Authors are suggested to consider the reference value which can be sufficient enough for such studies.
- How manufacturing process variables and real-time manufacturing activities related variables are linked with each other? Please justify.
- The graphs provided under S1 are not much cleared.
- Why BOD is not considered in this study under monitoring parameter?
- In this study, methodology is not much clear and must be described in detail. Section 3 must be placed after description of methodology which is mentioned in section 5.
- Most of the results of this study are not supported with references.
- Authors should provide the details of activation function, transfer function, and model performance parameters such as MSE, SSE, AAD, and MAE, along with R^2 and RMSE.
- Some of tables of SI can be shift to MS. Authors are suggested to condense this information.

Reviewer #2 (Remarks to the Author):

In general

In this paper, the author aims to establish a numerical relationship between the outflow of industrial by-products through the aqueous and solid phases. This relationship can be helpful in developing a low-cost and efficient method for evaluating three waste discharge. The reliability of the model results is enhanced by utilizing the random forest model and comparing it with other algorithms. The large sample size of the training set also provides a solid foundation for the emergence of model mechanisms. However, it should be noted that this study is based on a regional survey. To increase the impact of this paper, three aspects need to be addressed:

i) It is important to provide prior knowledge of supervised learning models, including relevant literature or reasoning. Additionally, explaining the causal relationships between independent and dependent variables through probabilistic models, such as Bayes, would enable future researchers to investigate their cases reliably using the methods proposed in this paper.

For example, the evaluation of waste generation requires an assessment of production efficiency, along with supporting data on material conservation, economic equivalence, atomic utilization rate, recovery rate, and raw material conversion rate. However, determining the level of harm caused by waste, considering factors such as toxicity, flammability, corrosiveness, explosiveness, degradability, radioactivity, and biological damage, necessitates data and judgment based on the nature of the waste. The methodology for obtaining this evidence should be further discussed.

ii) Since this study is not a global survey, it is crucial to consider regional characteristics. For instance, it would be valuable to analyze whether the proportion of by-products output by industrial and mining enterprises through the water or solid phase has changed in areas with varying water consumption levels.

iii) Furthermore, it is essential to examine whether there are differences in the characteristics of the industrial sectors surveyed. If the industrial layout and research area differ in other regions, appropriate corrections should be made to account for these variations.

This article presents the application of a machine learning algorithm to derive additional environmental variables from known ones. While the methodology appears to be well-developed, further exploration is required to assess the significance of its environmental impact assessment and its potential for broader applications. I suggest publishing it after making major revisions.

In detail

Line 10 In abstract, the clarity of the logical representation of the principle innovation, technical significance, and process proof is important. The text should focus on the innovation of the work, that is, the most important environmental effects or methodological breakthroughs in environmental assessment learned from the data set.

Line 35 In the five paragraphs of the introduction, the logical assumptions and expressions promoted by the problem are not currently clear enough. The most important scientific innovations and research plans need to focus on material, production, logistics, and management aspects, which require further clarification. Throughout the discussions, there is little reference to the characteristics of wastewater, the correlation between wastewater and waste, and the supporting relationship of discharge laws.

Line 75 As a transitional paragraph in the introduction, it is important to explain the connection between the quantity of sewage and hazardous waste, including any previous research or discoveries on this topic. Furthermore, it is necessary to discuss the feasibility of using sewage as a measure for hazardous waste, the challenges in comparing sewage and hazardous waste evaluations, and whether the approach of "obtaining hazardous waste information through water analysis" is adopted due to the cost-effectiveness and efficiency of sewage detection and monitoring methods.

Line 112 In machine learning, feature engineering plays a crucial role. Wastewater and solid waste (HW) discharge data need to be evaluated in conjunction with production data, and proof of the reliability of statistical data sources needs to be adequately provided. There is insufficient evidence for the rules of variable screening. On one hand, variables could be screened based on prior knowledge, and their selection could be guided by necessary indicators in engineering experience and chemical engineering literature. This process can be referred to as a preselection process. On the other hand, it is important to verify the causal relationship between independent variables and dependent variables

using actual data performance. This can be done by employing probabilistic models such as Bayes formula and analyzing sensitivity. Non-causal variables should be excluded to avoid any misleading results. Additionally, a simplified approach can be used to replace multiple variables with a single variable. While these processes can be described in the supporting materials, the screening results should be presented in the manuscript.

Line 137 The allocation of water and solids in the species association should be a phase allocation process at each stage of the process and should therefore be tailored to the elemental composition of hazardous wastes. However, since the types of hazardous wastes involved are not clearly defined, it is challenging to determine if the 12 water variables can fully describe all hazardous wastes.

Line 182 The size of RSME should refer to the mean value, which should be labeled or indicated in Figure 3.

Line 201 Does the performance optimization here compare to other interpolation methods?

Line 211 Section 3.1 and Section 3.2 can be combined because the second half of 3.1 and 3.2 deals with the handling of model boundary values.

Line 254 For the study of Section 3.3 Sector-independent model development, it is suggested to use the analysis of three cases to compare with others, emphasizing the importance and universal practicability of the method.

Line 298 In the discussion of Section 3.4 Influence of input features on predicting hazardous waste generation, there is a lack of support in terms of material nature, production technology level, management level and manpower level.

Line 308 Before determining the importance of features, please make sure that the 18 variables of the industrial sector, the firm scale, and manufacturing processes are independent.

Line 380 Hopefully, there is a summary group diagram to express the results.

Line 414 In the writing of Policy Implication, it is necessary to strengthen the expression of actionable and enforceable ideas and highlight the technical significance of innovation. There are two points that require further examination. Firstly, is there a correlation between start-up/retrofit time and the model? In other words, does the advanced nature of technology affect the proportion of solid liquid pollution? Secondly, can regional water conservation policies impact the results of the model? Since the study area encompasses multiple watersheds, water prices and availability are not constant.

Reviewer #3 (Remarks to the Author):

In this paper, the authors utilized real-time wastewater data and static data to conduct firm-level estimation of hazardous waste generation. The execution of this study has been comprehensive, yielding some useful conclusions for waste management. However, the innovativeness of this study is somewhat lacking, particularly in its methodology, where the proposed methods are the combinations of common traditional machine learning algorithms. Moreover, the outcomes yielded by the proposed methods are less than satisfactory, with R2 values of about 0.8, which implies that the proposed methods lack practicality. This work may be interesting for in hazardous waste management, but it may be difficult to generate interest in the broader research fields. Some other detailed comments are listed below for the authors' reference.

(1) The author should engage in a more comprehensive discussion regarding the proposed scheme. From an enterprise perspective, does this scheme bear the potential to augment the company's expenditures? From a managerial standpoint, can this scheme yield dependable outcomes? And how cost-effective is the proposed scheme compared to other management measures?

(2) As the authors have mentioned, "In the environmental management, lacking the ground-truth knowledge is the most intractable challenge, ...", how can one ascertain the veracity of the data declared by the enterprises? If the data declared by the enterprises is unreliable, the prediction results of the proposed methods will be even more unreliable.

(3) Depending on the accuracy of the model (R2 values are about 0.8), can the methods address the waste management problems mentioned by the authors?

(4) It is suggested that the author incorporate repetitive experiments to elucidate the reliability of the

predictive outcomes.

(5) What is the number of missing values being filled in? Does the number of filled values affect the reliability of the model outcomes?

(6) The author should elucidate the rationale behind employing staff number as a representation of a firm scale, given the varying degrees of labor intensity across different industries.

Itemized Replies to Review Comments

(Manuscript Number: NCOMMS-23-45555)

Comments by reviewer: 1

Overview:

The study is done thoroughly and can have relevance for industrial stakeholders and other policy makers. I have few doubts about this MS, please revise as per the given comments/suggestion.

Authors' response:

We would like to thank the reviewer for the positive feedback on the significance of this study. Also, revision based on your comments point-by-point has been made to further improve the quality of this paper. Below are point-to-point responses in which we explicitly refer to the relevant lines in the revised manuscript/supplementary information to clarify the genuine. The comments appear in black, our responses appear in indent and blue font, and the texts quoted from the revised manuscript/supplementary information appear in italics and green font.

Specific comments:

1. The introduction section is too lengthy, readers may lose interest while going through this part. Please condense it remove the redundant information.

Authors' response:

Suggestion taken. Combining reviewer #2's suggestion to clarify the logical assumption of linking wastewater data to source estimation of hazardous waste (HW) as well as the advancement of adopting this data-driven approach in management practice, we have condensed the introduction by eliminating unnecessary details, ensuring a more concise and engaging presentation for readers.

The reorganized introduction consisted of four key parts. Firstly, we highlighted the significance of HW management and elaborated the major challenge faced in HW management is the accurate source estimation of HW generation at the firm-level. Particularly for enormous enterprises with high heterogeneity, an

efficient method to estimate the HW generation which could obtain a detailed and conducive data at the firm-level is lacked.

Secondly, we delineated the existing gaps in current method to estimate the HW generation from enterprise. These gaps primarily centered around the lack of ground-truth knowledge regarding the pattern of HW generation for each enterprise^{1,2} and the absence of production activity data with sufficient temporal granularity to support HW source estimation during management practices^{3,4}.

Thirdly, in response to these research gaps, we put forward a novel idea of utilizing the big data of wastewater to predict HW generation for each enterprise based on the machine learning algorithms. The concept is grounded in the availability of wastewater big data with application of Internet of Things (IoT) sensors in environmental management, which could depict the behavior of heterogenous enterprises⁵, and the logical assumption that a correlation exists between wastewater and HW generation. This correlation is plausible as both of wastewater and HW generation are directly linked to manufacturing processes, where the generation of wastewater and HW results from the partition of contaminants in liquid and solid phases^{6,7}. Then, we stressed that the advancement of this data-driven approach in management practice existed in two aspects: (a) the high granular time resolution of wastewater data, accessible to environmental stakeholders, supporting higher frequency updates for HW source estimation, and (b) the cost-effectiveness in terms of time and labor compared to traditional field survey methods.

Fourthly, we proposed a generic framework that jointly used representative variables from different industrial sectors, exploited data-balance algorithm to address long-tail data distribution, and incorporated causal discovery to screen features and improve computation efficiency. The feasibility of this framework was demonstrated through an application using 4,260,593 daily wastewater emission data to predict HW generation from 1024 enterprises across 10 industrial sectors in Jiangsu, one of the most industrialized provinces in China.

For more detailed information about the revised Introduction, please see texts in lines 38–108 of the revised manuscript.

2. The number of variables considered in this study for enterprise characteristics and real time manufacturing process may not be sufficient for model validation. Authors are suggested to consider the reference value which can be sufficient enough for such studies.

Authors' response:

Suggestion taken. To ensure the inclusion of sufficient variables for model development, we have extended the number of variables to 43 in the revised manuscript to enhance its generalizability to most industrial sectors and HW categories. Such modification is grounded in a detailed investigation on the HW and wastewater generation characteristics and manufacturing processes for 50 typical industrial sectors (See Supplementary Table 3 & 4) through the literature review and national/industrial guideline/regulations. These 50 sectors, classified based on the 4-digit National Standard Industrial Classification, contributed to 94.2% of the total HW generation quantity in China in 2015 according to the China Environmental Statistics Database.

Specifically, variables in this study were selected from two perspectives of static enterprise characteristics and real-time manufacturing activities because previous studies have demonstrated that HW generation shows strong affinities with both of these two factors^{8,9}. Since the aim of this study is to build a uniform as well as practical data-driven approach achieving accurate source estimation of HW, the selection of variables was based on the principle of not only complete but also availability for environmental regulatory. Thus, in this generic model framework, sector, manufacturing processes, and firm scale were used to represent the static enterprise characteristics. Regarding the features about real-time manufacturing activities, we have omitted some variables, such as raw and auxiliary material inputs and product outputs, because acquiring high-resolution data for these features remains challenging. In contrast, we used wastewater discharge to represent real-time manufacturing activities because wastewater, as by-product¹⁰, could not only indicate manufacturing activities, but also has been demonstrated to have relationship with HW generation according to a thorough literature review of 71 papers related to chemical and engineering experience.

Overall, this generic framework involved 43 variables (19 variables related to enterprise characteristics and 24 variables related to real-time manufacturing

activities). These 19 variables related to enterprise static characteristics encompassed the industrial sector that the firm belongs to, the firm scale (See Supplementary Table 2), and 17 manufacturing processes (See Supplementary Table 3). Features of 17 manufacturing processes were binary variables which were determined as 1 if the firm has this manufacturing process; otherwise, the values were 0. These processes were summarized according to the Chinese Environmental Management Guideline for Hazardous Wastes and literature review.

24 variables used to indicate the firm's real-time activities were features of water contaminate emission, consisting of wastewater discharge amount, chemical oxygen demand (COD), pH, ammonia nitrogen (NH₃-N), total nitrogen (N), total phosphorus (P), suspended substance, petroleum hydrocarbon pollutants, biochemical oxygen demand, volatile phenol, total organic carbon, sulphate, fluoride, cyanide, lead, arsenic, cadmium, mercury, iron (Fe), total chromium (Cr), hexavalent chromium (Cr^{VI}), copper (Cu), zinc (Zn), and nickel (Ni). These 24 indicators were mandated by the environmental regulatory to be monitored automatically or manually based on national or industrial regulations. They showed tight affinities with most of HW categories according to engineering and chemical experiences (See Supplementary Table 1).

For more detailed information about the variable selection, please see texts in lines 109–156 of the revised manuscript.

Then, under this generic framework, for specific cases, features could be screened from these 43 variables and tailored based on characteristics of studied sectors and the data availability. With respect to the case (1024 enterprises from 10 industrial sectors with the highest HW generation quantity in Jiangsu Province, China) we studied, 25 variables, including the industrial sector that the firm belongs to, the firm scale, 11 manufacturing processes, and 12 variables of wastewater monitoring indicators (wastewater discharge amount, COD, pH, NH₃-N, N, P, Fe, Cr, Cr^{VI}, Cu, Zn, and Ni), were screened for model developments. Results showed that using these variables, the combined model to predict the generation quantity of HW achieved a satisfactory performance with R² of 0.87, which is much higher than most other models to predict HW generation^{3,11}. In addition, we have added uncertainty analysis in the revised

manuscript and results showed that for the combined models to predict generation quantity of HW, 95.85% testing samples fell into the 95% confidence intervals of prediction, indicating that the prediction value was reliability for management practice. Therefore, we believe that although additional variables might enhance model performance, the current set is considered sufficient.

Even so, we admit that involving more variables might further improve the model performance. In the section of 4.4 Limitation and future research, we have strengthened that this study aims to develop a generic and flexible model framework. If possible, more variables, such as electricity consumption data, encompassing enterprises' static characteristic and real-time manufacturing activities could be included in the model, but the model development scheme is same as the studied cases. The statement of this perspective can be found in lines 582–589 of the revised manuscript.

3. How manufacturing process variables and real-time manufacturing activities related variables are linked with each other? Please justify.

Authors' response:

Suggestion taken. We have added the description about the data structure to show linkage between enterprise' static characteristics and real-time manufacturing activities in the revised supplementary information. Specifically, the 16,300 observations used to build model after data processing were from 1024 enterprises, which means that each enterprise had about 16 observations on average. For these observations from the same enterprise, their value of enterprise' static characteristics, such as firm scale and manufacturing processes, kept constant, but the data of real-time manufacturing activities, mainly represented by the wastewater data, varied across observations, reflecting the change in enterprise's behavior at different sampling times. The newly added description could be found in lines 96–103 of the revised supplementary information.

It is true that a specific manufacturing process might lead to the discharge of particular contaminants in wastewater¹², but such intricate linkage cannot be reflected in the available data because the one observation described the overall behavior and characteristics of an enterprise throughout the observed month, making it challenging to pinpoint specific contaminant discharge amount to

individual processes.

4. The graphs provided under S1 are not much cleared.

Authors' response:

Suggestion taken. In the revised manuscript, we have enhanced the resolution of all figures to 600 dpi throughout the manuscript.

5. Why BOD is not considered in this study under monitoring parameter?

Authors' response:

Suggestion taken. In the revised manuscript, we have involved the wastewater indicator of biochemical oxygen demand (BOD) as a variable in the generic framework. Please see texts in lines 140–147 of the revised manuscript for more details.

Specifically, to put forward a more generic framework being applicable to most industrial sectors and HW categories, we have investigated the typical pollutants in wastewater and its relationship with HW generation for 50 typical industrial sectors (See Supplementary Table 1 & 4). Based on literature review from 71 papers and 25 Chinese national or industrial regulations, 24 indicators (wastewater discharge amount, chemical oxygen demand, pH, ammonia nitrogen, total nitrogen, total phosphorus, suspended substance, petroleum hydrocarbon pollutants, biochemical oxygen demand, volatile phenol, total organic carbon, sulphate, fluoride, cyanide, lead, arsenic, cadmium, mercury, iron, total chromium, hexavalent chromium, copper, zinc, and nickel) about wastewater were involved in the generic framework.

However, in the studied case, the variable of BOD was not included since it is not mandated to be monitored by Internet of Things (IoT) sensors according to Administration Regulation for Automatic Monitoring of Pollution Discharge in Jiangsu Province¹³, so there was no BOD data for model developments. If the BOD data become available in the future, it could be involved in input features to further improve the model predictive performance.

6. In this study, methodology is not much clear and must be described in detail. Section 3 must be placed after description of methodology which is mentioned in section 5.

Authors' response:

Sorry for the confusion. We have made a clearer description of methodology, including the variable selection, feature engineering, model development, and uncertainty analysis, in the revised manuscript. The Figure 2 (see below) of model framework has been revised to help readers gain a comprehensive understanding of the methodology. Besides, additional detailed information of the methodology, such as mission value imputation, data source and quality control, outlier rejection, and metrics to evaluate the performance of models, has been presented in the revised supplementary information.

[redacted]

Figure 2. The model framework for predicting the total generation quantity of HW and one specific category. (a) Data of variables which could indicate static characteristics and real-time activities of enterprises are collected and

preprocessed. (b) Feature engineering which incorporated the causal discovery, importance, and correlation analysis was conducted to screen input features. (c) A regression model is developed directly from the training dataset with data balance to predict the total generation quantity of HW. (d) An ensemble model coupling the classification and regression model is developed to predict the generation quantity of one category of HW. The binary classification model could determine whether the generation quantity of this category is 0. The regression model is developed to predict the specific value when the generation quantity of this category was > 0 based on the classification model results. (e) After model development, performance validation is conducted on the test dataset. Additionally, feature exploration is performed on the trained models to investigate the impact of input features on the target output.

Specifically, the selection of variables has been clearly explained by initially introducing a scheme of variables selection from two perspectives of static enterprise characteristics and real-time manufacturing activities because previous studies have demonstrated that HW generation shows strong affinities with both of these two aspects^{8,9}. Then, under this scheme, following the principle of data availability and the sufficiency to support model development, specific variables were determined based on the detailed investigation on HW generation, wastewater discharge, and manufacturing processes from 50 representative sectors. Please see texts in lines 109–193 of the revised manuscript for more details.

Secondly, in the Method section, we introduced three major parts of feature engineering, model development, and uncertainty analysis. The part of feature engineering stated how to screen useful and independent input features through incorporating the causal discovery, importance ranking, and correlation analysis. The part of model developments described the approach of data balance and processes of constructing machine learning models for the sector-combined and sector-independent models. The part of uncertainty analysis outlined the principle and method of quantifying the uncertainty of random forest model using quantile regression forest. Please see texts in lines 590–730 of the revised manuscript for more details.

Regarding the layout of the Results section (Section 3) being placed before the Methods section (Section 5), it followed the formatting instruction of the journal.

Therefore, we did not re-organize the structure, but we hope that the revised methodology was clear enough to help reader having a better understanding about the manuscript.

7. Most of the results of this study are not supported with references.

Authors' response:

Thanks for your comments. We have cited more reference to support the methodology and results discussion throughout the manuscript. To be specific, we have made a thorough literature review, including 71 papers, to investigate the relationship between water contaminant emissions and HW generation (See Supplementary Table 1 & 4), thereby supporting the research hypothesis and variables selection. These literatures mainly talked about the contaminate generation from various industrial sectors and their manufacturing processes. In addition, 25 Chinese national or industrial regulations have been reviewed to summarize the typical wastewater pollutant and monitoring indicators for various sectors.

Besides, 11 additional references have been cited in the section of discussing features' importance on model predictive performance. These newly cited references mainly supported discussions about (a) impacts of firm scale and manufacturing processes on the HW generation and (b) the underlying driving factors influencing the quantification relationship between wastewater and HW generation quantities.

8. Authors should provide the details of activation function, transfer function, and model performance parameters such as MSE, SSE, AAD, and MAE, along with R² and RMSE.

Authors' response:

Suggestion taken. We have incorporated additional model performance parameters of mean absolute error (MAE), mean absolute percentage error (MAPE), median absolute deviation (MAD), mean square error (MSE), and sum of squares due to error (SSE), alongside R² and RMSE, to provide a more comprehensive evaluation of regression models. Performances of all the regression models were evaluated using these 7 metrics and displayed in tables or figures. Also, classification models were evaluated using metrics of accuracy,

recall, precision, and F1-score (F1). More detailed information about calculation equations of these metrics can be found in lines 193–226 of the revised supplementary information.

Regarding the activation function and transfer function, we want to clarify that our models were based on the Random Forest algorithm, which does not involve specific activation or transfer functions¹⁴. These functions are typically associated with neural networks^{15,16}, but our approach utilized a different type of algorithms based on decision trees.

9. Some of tables of SI can be shift to MS. Authors are suggested to condense this information.

Authors’ response:

Suggestion taken. following the principle of presenting key findings in the form of tables or figures in the manuscript, we have added a table showcasing the model performances of three application cases in other regions or sectors. This addition aims to underscore the generalizability of our model framework. Notably, other significant results, such as the performance of combined and sector-independent models, as well as the importance of input features, have already been illustrated in Figures 3-6.

For the editor and reviewers’ convenience, we quote the newly added Table 1 as follows:

Table 1. Model performances of three application cases in other regions or sectors.

Region	Sector	R²	RMSE	Final Input Features
Zhejiang, China	metal surface treatment	0.72	721.15	manufacturing processes (including: metal surface processing, circuit board treatment, equipment maintenance, exhaust gas treatment, and wastewater treatment), firm scale, wastewater discharge amount, N, P, NH₃-N
Shandong, China	metal surface treatment	0.69	29.38	manufacturing processes (including: metal heat treatment, metal surface processing, circuit

Region	Sector	R²	RMSE	Final Input Features
				board treatment, equipment maintenance, exhaust gas treatment, and wastewater treatment), firm scale, COD, NH₃-N, N, Cr^{VI}, Cr
Hunan, China	lead and zinc metallurgy	0.82	2366.63	manufacturing processes (including: mineral leaching, electrolysis, roasting, and flue gas treatment), firm scale, COD, NH₃-N, P, Cr^{VI}

Comments by reviewer: 2

Overview:

In this paper, the author aims to establish a numerical relationship between the outflow of industrial by-products through the aqueous and solid phases. This relationship can be helpful in developing a low-cost and efficient method for evaluating three waste discharge. The reliability of the model results is enhanced by utilizing the random forest model and comparing it with other algorithms. The large sample size of the training set also provides a solid foundation for the emergence of model mechanisms. However, it should be noted that this study is based on a regional survey. To increase the impact of this paper, three aspects need to be addressed:

Authors' response:

We thank the reviewer very much for your agreement on the novelty of this work as well as the insightful and helpful suggestions. In response to the major limitation associated with this regional survey, we have addressed it by developing a comprehensive generic model framework and demonstrating its generalizability across various regions and industrial sectors, see below.

(1) It is important to provide prior knowledge of supervised learning models, including relevant literature or reasoning. Additionally, explaining the causal relationships between independent and dependent variables through probabilistic models, such as Bayes, would enable future researchers to investigate their cases reliably using the methods proposed in this paper. For example, the evaluation of waste generation requires an assessment of production efficiency, along with supporting data on material conservation, economic equivalence, atomic utilization rate, recovery rate, and raw material conversion rate. However, determining the level of harm caused by waste, considering factors such as toxicity, flammability, corrosiveness, explosiveness, degradability, radioactivity, and biological damage, necessitates data and judgment based on the nature of the waste. The methodology for obtaining this evidence should be further discussed.

Authors' response:

We have addressed the concern regarding the lack of sufficient prior knowledge during variable selection by conducting a comprehensive review on literature and national/industrial regulations. Additional 71 papers about chemical and engineering experience and 25 national/industrial guideline/regulations about

manufacturing processes and contaminant monitoring of most industrial sectors, have been reviewed to comprehensively collecting the evidence underlying adopting wastewater data to predict HW generation, which is used as the prior knowledge to support the selection of variables.

Based on the prior knowledge, in the revised manuscript, we extended the number of variables into 43 (19 variables related to enterprise characteristics and 24 variables related to real-time manufacturing activities) to make it generic for most scenarios.

These 19 variables related to enterprise static characteristics encompassed the industrial sector that the firm belongs to, the firm scale (See Supplementary Table 2), and 17 manufacturing processes (See Supplementary Table 3). 24 variables reflecting the firm's real-time activities were features of water contaminate emission, consisting of wastewater discharge amount, chemical oxygen demand, pH, ammonia nitrogen, total nitrogen, total phosphorus, suspended substance, petroleum hydrocarbon pollutants, biochemical oxygen demand, volatile phenol, total organic carbon, sulphate, fluoride, cyanide, lead, arsenic, cadmium, mercury, iron, total chromium, hexavalent chromium, copper, zinc, and nickel. For specific cases, features could be screened from these 43 variables and tailored based on characteristics of studied sectors and data availability.

Then, to ensure that the input features are independent and reasonable, the procedure of feature screening incorporating causal relationship analysis, importance ranking, and correlation analysis was newly developed and involved in the framework. Firstly, the causal relationship was analyzed via directed acyclic graph (DAG) learning, where we derived a set of cause-and-effect relationships from the observational data¹⁷, and the minimal set of features with high relevance to the response variable was obtained based on the Markov Blanket (MB) search algorithm¹⁸. Subsequently, features with high importance on model prediction but not being identified as relevance features were added, while those with low importance but high correlation were eliminated.

Furthermore, we conducted an in-depth evaluation of the effect of different feature selection principles, including (i) screening features based on MB search algorithm according to the DAG learning and (ii) combining the features

screened based on MB search algorithm, features' importance ranking, and correlation analysis, on the model performance. The comparison showed that relying solely on MB-based features sometimes led to a decrease in predictive performance (See Supplementary Table 11 & 12). This reduction was because the MB learning algorithm might fail to identify some critical features during feature selection tasks caused by the strict assumption of data distribution, variable types, or correctness of criteria, during causal discovery, which thereby limited the faithfulness of DAG learning results^{19,20}. However, when the selected features were further refined based on features' importance and correlation analysis, the model even could achieve superior performance to the baseline models with all variables as input. This robust validation supported the rationality of our newly developed feature engineering method that incorporated the causal discovery, importance ranking, and correlation analysis.

(2) Since this study is not a global survey, it is crucial to consider regional characteristics. For instance, it would be valuable to analyze whether the proportion of by-products output by industrial and mining enterprises through the water or solid phase has changed in areas with varying water consumption levels.

Authors' response:

To overcome the limitation of the regional study caused by the data availability, we have made a more detailed investigation on characteristics of HW generation, wastewater discharge, and manufacturing processes from most industrial sectors and then put forward a generic model framework will could be feasible in other regions and industrial sectors. The generalizability of this framework was demonstrated by developing cases application considering the regional and sectoral differences.

As the reviewer stated, a key regional difference lies in the varied partition ratio of contaminants between solid and liquid phases, which might be influenced by regional water withdrawal strategy and technological advancement²¹. Take the sector of metal surface treatment (MST) for example. The ratio of wastewater discharge amount to HW generation quantity was used to indicates different contaminant partition relationship between solid and liquid phases. The Kruskal-Wallis Test based on the data from 2,727 enterprises across China in 2015 showed that even for the same sector, some regions displayed significant

difference in the contaminant partition ratio between liquid and solid phase ($p < 2.2e^{-16}$)²².

Then, in response to this pattern, two cases were crafted to illustrate the model's applicability in regions with varying contaminant partition between liquid and solid phases. Specifically, the first application focused on the development of model for sector of MST in Shandong province which had lower ratio of WTH than the studied region of Jiangsu. The sector of MST was chosen because it had sufficient sample size to build models for most provinces. The second focuses on model development in Zhejiang province which had higher WTH than the studied region of Jiangsu.

For the two cases developed for regions with different WTH ratios, models presented commendable predictive performances (Shandong province: $R^2=0.69$, $RMSE=29.38$; Zhejiang province: $R^2=0.72$, $RMSE=721.15$). This suggested that our model framework adeptly captured and simulated the heterogeneous relationship between wastewater and HW generation, even when it varied across regions. In addition, in the Zhejiang case, the model developed for this region was applied to predict HW generation in Shanghai, a region in the same cluster as Zhejiang based on WTH ratio. However, the prediction performance was below expectations ($R^2=0.32$). This implied that the model must be trained using the localized data derived from the application region.

(3) Furthermore, it is essential to examine whether there are differences in the characteristics of the industrial sectors surveyed. If the industrial layout and research area differ in other regions, appropriate corrections should be made to account for these variations.

Authors' response:

Another key regional difference is the varied industrial sector layout, which means that the representative industrial sectors generating HW changed with regions²³. Aiming at this issue, we have made a detailed investigation on industrial sector layout for different provinces in China to reveal the major industrial sectors generating HW (See Supplementary Table 4), which thereby provided a basis for the framework application in different regions.

Based on the investigation on the regional characteristics of industrial sector

layout, we chose another representative industrial sector in a different region to demonstrate how our generic model framework can be used to predict the HW generation in varied industrial sectors. This case focused on the industrial sector of lead and zinc metallurgy sector in Hunan province, a sector not studied in Jiangsu but significant for HW generation in China. The key adaptation during developing this case was adjusting the variables of manufacturing processes based on the characteristics of the studied sector. Results from this case revealed that based on this adaptation, the model still exhibited high reliability and prediction accuracy with the R^2 and RMSE values of 0.82 and 2366.63, respectively (See Table 1).

The above cases to demonstrate the generalizability of the framework reaffirmed the potential of our framework for application in diverse regions and industrial sectors, provided that model training data is localized, and variables related to manufacturing processes are appropriately screened based on the specific studied sector.

This article presents the application of a machine learning algorithm to derive additional environmental variables from known ones. While the methodology appears to be well-developed, further exploration is required to assess the significance of its environmental impact assessment and its potential for broader applications. I suggest publishing it after making **major revisions**.

Authors' response:

We would like to thank the reviewer for the positive feedback on the novelty of the methodology in this study. We have carefully prepared this revised version. We believe that the reviewers' suggestions together with our input have greatly improved this paper.

Below are point-to-point responses in which we explicitly refer to the relevant lines in the revised manuscript/supplementary information to clarify the genuine. The comments appear in black, our responses appear in indent and blue font, and the texts quoted from the revised manuscript/supplementary information appear in italics and green font.

Specific comments:

1. **Line 10** In abstract, the clarity of the logical representation of the principle innovation, technical significance, and process proof is important. The text should focus on the innovation of the work, that is, the most important environmental effects or methodological breakthroughs in environmental assessment learned from the data set.

Authors' response:

Suggestion taken. We have carefully revised the abstract to stress the principal innovation, technical significance, and process proof. Firstly, we highlighted the significance of tracking contaminant generation at the firm-level and elaborated the major challenge of lacking a universally applicable estimation method for heterogeneous enterprises. Secondly, we stated the reason of taking HW as a case and the rationale of the innovative idea of linking wastewater data, which could be automatically monitored, to predict HW generation. Thirdly, based on the idea, we proposed a data-driven and generic approach with the technical innovation of designing a generic framework that jointly used representative variables from different industrial sectors, exploited data-balance algorithm to address long-tail data distribution, and incorporated causal discovery to screen features and improve computation efficiency. Finally, the feasibility of the model framework was demonstrated through a successful application to 1024 enterprises across 10 industrial sectors in Jiangsu, one of the most industrialized provinces in China.

For the editor and reviewers' convenience, we quote the revised abstract (Line 16–35 in the revised manuscript) as follows:

Industrial enterprises are prominent sources of contaminant discharge on the planet and regulating their operations is vital for sustainable development. However, accurately tracking contaminant generation at the firm-level remains an intractable global issue due to significant heterogeneities among enormous enterprises and the absence of a universally applicable estimation method. This study addressed the challenge by focusing on hazardous waste (HW), known for its severe harmful properties and difficulty in automatic monitoring, and developed a data-driven methodology that predicted HW generation utilizing wastewater big data in a uniform and lightweight manner. The idea is grounded in the availability of wastewater big data with widespread application of automatic sensors, enabling depiction of heterogeneous enterprises, and the logical assumption that a correlation exists between wastewater and HW

generation. We simulated this relationship by designing a generic framework that jointly used representative variables from diverse sectors, exploited a data-balance algorithm to address long-tail data distribution, and incorporated causal discovery to screen features and improve computation efficiency. To illustrate our approach, we applied it to 1024 enterprises across 10 sectors in Jiangsu, a highly industrialized province in China. Validation results demonstrated the model's high fidelity ($R^2=0.87$) in predicting HW generation using 4,260,593 daily wastewater data.

2. **Line 35** In the five paragraphs of the introduction, the logical assumptions and expressions promoted by the problem are not currently clear enough. The most important scientific innovations and research plans need to focus on material, production, logistics, and management aspects, which require further clarification. Throughout the discussions, there is little reference to the characteristics of wastewater, the correlation between wastewater and waste, and the supporting relationship of discharge laws.

Authors' response:

Suggestion taken. Firstly, in the revised manuscript, to support the basic assumption that wastewater big data could deliver new knowledge about HW generation, we have conducted a thorough literature review, including 71 papers, to collect the chemical and engineering evidence about the relationship between wastewater and HW generation (See Supplementary Table 1).

Secondly, to make a clearer introduction, combining the reviewer #1's suggestion to condense the introduction and eliminate unnecessary details, we have reorganized the section of Introduction and centered on clarifying the logical assumption of linking wastewater data to source estimation of HW and the advancement of adopting this data-driven approach in management practice.

The reorganized introduction consisted of four key parts. In the first part, we highlighted the significance of HW management and elaborated the major challenge faced in HW management is the accurate source estimation of HW generation at the firm-level. Particularly for enormous enterprises with high heterogeneity, an efficient method to estimate the HW generation which could obtain a detailed and conducive data at the firm-level is lacked.

In the second part, we stated the existing gaps in current method to estimate the HW generation from enterprise. These gaps primarily centered around the lack of ground-truth knowledge regarding the patterns of HW generation for each enterprise^{1,2} and the absence of production activity data with sufficient temporal granularity to support HW source estimation during management practices^{3,4}.

In the third part, in response to these research gaps, we put forward a novel idea of utilizing the wastewater big data to predict HW generation for each enterprise based on the machine learning algorithms. The concept is grounded in the availability of wastewater big data with application of Internet of Things (IoT) sensors in environmental management, to depict the behavior of heterogeneous enterprises⁵, and the logical assumption that a correlation exists between wastewater and HW generation. This correlation is plausible as both of wastewater and HW generation are directly linked to manufacturing processes, where the wastewater and HW generation results from the partition of contaminants in liquid and solid phases, respectively^{6,7}. Then, we stressed that the advancement of this data-driven approach in management practice existed in two aspects: (a) the high granular time resolution of wastewater data, accessible to environmental stakeholders, supporting higher frequency updates for HW source estimation, and (b) the cost-effectiveness in terms of time and labor compared to traditional field survey methods.

In the fourth part, we proposed a generic framework that jointly used representative variables from different industrial sectors, exploited data-balance algorithm to address long-tail data distribution, and incorporated causal discovery to screen features and improve computation efficiency. The feasibility of this framework was demonstrated through an application using 4,260,593 daily wastewater emission data to predict HW generation from 1024 enterprises across 10 industrial sectors in Jiangsu, one of the most industrialized provinces in China.

For more detailed information about the revised Introduction, please see texts in lines 38–108 of the revised manuscript.

3. **Line 75** As a transitional paragraph in the introduction, it is important to explain the connection between the quantity of sewage and hazardous waste, including any previous research or discoveries on this topic. Furthermore, it is necessary to discuss

the feasibility of using sewage as a measure for hazardous waste, the challenges in comparing sewage and hazardous waste evaluations, and whether the approach of "obtaining hazardous waste information through water analysis" is adopted due to the cost-effectiveness and efficiency of sewage detection and monitoring methods.

Authors' response:

Suggestion taken. In the revised section of Introduction, we have organized two paragraphs to clearly elaborate the research gap and our contribution, the logical assumption of existing a correlation between wastewater and HW generation, and the advancement of this approach in the real-world management practice.

In the first paragraph, we elaborated the existing gaps in current method to estimate the HW generation from enterprise. These gaps primarily centered around the lack of ground-truth knowledge regarding the patterns of HW generation for each enterprise^{1,2} and the absence of production activity data with sufficient temporal granularity to support HW source estimation during management practices^{3,4}.

In the second paragraph, in response to the research gaps, we put forward a novel idea of utilizing the wastewater big data to predict HW generation for each enterprise based on the machine learning algorithms. The concept is grounded in the availability of wastewater big data with application of Internet of Things (IoT) sensors in environmental management, to depict the behavior of heterogenous enterprises⁵, and the logical assumption that a correlation exists between wastewater and HW generation. This correlation is plausible as both of wastewater and HW generation are directly linked to manufacturing processes, where the generation of wastewater and HW results from the partition of contaminants in liquid and solid phases, respectively^{6,7} (Detailed evidences could be found in Supplementary Table 1). Then, we stressed that the advancement of this data-driven approach in management practice existed in two aspects: (a) the high granular time resolution of wastewater data, accessible to environmental stakeholders, supporting higher frequency updates for HW source estimation, and (b) the cost-effectiveness in terms of time and labor compared to traditional field survey methods. Please see texts in lines 65–94 of the revised manuscript for more details

4. **Line 112** In machine learning, feature engineering plays a crucial role. Wastewater and solid waste (HW) discharge data need to be evaluated in conjunction with production data, and proof of the reliability of statistical data sources needs to be adequately provided.

Authors' response:

Suggestion taken. We have made a detailed description of the data source and data processing to ensure the reliability of our data for model development.

Firstly, to ensure the credibility of the HW generation data declared by enterprises, we meticulously eliminated data from enterprises with records of environmental violations and administrative penalties. These records of environmental enforcement were sourced officially from the Department of Ecology and Environment of Jiangsu Province, China. Please see texts in lines 46–66 of the revised supplementary information for more details.

Then, after data aggregation, we implemented a robust outlier screening and removal procedure, incorporating six unsupervised machine learning algorithms: K Nearest Neighbors, Minimum Covariance Determinant, Clustering-Based Local Outlier Factor, Histogram-based Outlier Score, Local Outlier Factor, and Isolation Forest. To minimize bias in outlier detection, the results from these algorithms were aggregated using the Average of Maximum (AOM) strategy²⁴. Please see texts in lines 68–94 of the revised supplementary information for more details.

Furthermore, to demonstrate the reliability of the data, we have conducted a statistical analysis on the relationship between HW generation, wastewater discharge amount, and enterprise characteristics. Due to the absence of detailed data on enterprise characteristics, such as the economic/physical output, we employed the firm scale, which has been widely used as an indicator of manufacturing activities, as a representative indicator of enterprises characteristics. The results consistently revealed a positive correlation, indicating a rising trend in both HW generation quantity and wastewater discharge amount with firm scale across all 10 industrial sectors in the studied case (Table R1), which is consistent with the common sense and partly supports the reliability of the data from the statistical perspective.

Table R1. The statistical description of the change of hazardous waste generation and wastewater discharge amount with firm scale

Industrial sector	Firm scale	Average hazardous waste generation (t)	Average wastewater discharge amount (t)
OCM	I	9.72	1917.32
	II	41.42	3415.48
	III	168.25	9007.05
	IV	380.83	23343.84
	V	843.76	52887.93
CPM	I	4.10	1871.99
	II	84.40	1186.63
	III	343.14	3631.49
	IV	466.10	12377.96
	V	901.78	31496.70
SCP	I	104.88	1474.97
	II	158.58	2968.11
	III	308.92	3793.92
	IV	1314.49	8263.34
	V	564.80	20594.26
STE	V	1042.39	540204.95
SRP	I	69.43	377.08
	II	76.81	2292.18
	III	147.37	8370.99
	IV	666.86	9110.14
	V	1278.68	44661.25
MWR	I	31.68	1951.08
	II	94.45	3840.51
	III	189.77	3509.23
	IV	245.42	7731.71
	V	578.27	12184.83
MST	I	17.36	1018.74
	II	40.95	2115.09
	III	64.05	3670.08
	IV	91.98	6907.42
	V	83.08	8047.50
ECM	I	80.68	2508.56
	II	174.14	3875.15
	III	74.77	1486.47
	IV	308.36	14918.22
	V	1194.28	73880.05
BEG	I	0.17	27855.63
	II	1040.74	8540.00
	III	1132.36	8126.24

Industrial sector	Firm scale	Average hazardous waste generation (t)	Average wastewater discharge amount (t)
EGU	II	815.59	12410.09
	III	1109.48	17681.05

There is insufficient evidence for the rules of variable screening. On one hand, variables could be screened based on prior knowledge, and their selection could be guided by necessary indicators in engineering experience and chemical engineering literature. This process can be referred to as a preselection process.

Authors' response:

Suggestion taken. We have addressed the concern regarding the lack of sufficient prior knowledge during variable selection by conducting a comprehensive review on literature and national/industrial regulations.

Firstly, a thorough literature review, including 71 papers from previous chemical and engineering experience, has been conducted to investigate the relationship between wastewater and generation of 46 HW categories²⁵ (See Supplementary Table 1). These literatures mainly talked about the contaminant generation from various industrial sectors and their manufacturing processes. It supported the hypothesis that there must exist correlation between wastewater and HW generation data since both of them are directly related to manufacturing processes where the generation of wastewater and HW were resulted from the partition of contaminates in liquid and solid phases^{6,7}. Also, features related to wastewater indicators were determined combining this literature review and national or industrial regulations introducing the wastewater indicators that mandated to be monitored automatically or manually

Then, regarding the complex factor of manufacturing processes, we summarized the dominant manufacturing processes generating HW for 50 typical industrial sectors according to the Chinese Environmental Management Guideline for Hazardous Wastes and literature review. These 50 sectors, classified based on the 4-digit National Standard Industrial Classification, contributed to 94.2% of the total HW generation quantity in China in 2015 according to the China Environmental Statistics Database.

Overall, based on the comprehensive review on literature and national/industrial guideline/regulations, in the revised manuscript, we have extended the number variables to 43 from the two aspects of static enterprise characteristics and real-time manufacturing activities. There were 19 variables related to enterprise static characteristics, encompassing the industrial sector that the firm belongs to, the firm scale (See Supplementary Table 2), and 17 manufacturing processes (See Supplementary Table 3). 24 variables used to indicate the firm's real-time activities were features of water contaminate emission, consisting of wastewater discharge amount, chemical oxygen demand, pH, ammonia nitrogen, total nitrogen, total phosphorus, suspended substance, petroleum hydrocarbon pollutants, biochemical oxygen demand, volatile phenol, total organic carbon, sulphate, fluoride, cyanide, lead, arsenic, cadmium, mercury, iron, total chromium, hexavalent chromium, copper, zinc, and nickel. For specific cases, features could be screened from these 43 variables and tailored based on characteristics of studied sectors and data availability. Please see texts in lines 109–156 of the revised manuscript for more details.

On the other hand, it is important to verify the causal relationship between independent variables and dependent variables using actual data performance. This can be done by employing probabilistic models such as Bayes formula and analyzing sensitivity. Non-causal variables should be excluded to avoid any misleading results. Additionally, a simplified approach can be used to replace multiple variables with a single variable. While these processes can be described in the supporting materials, the screening results should be presented in the manuscript.

Authors' response:

Suggestion taken. In the revised manuscript, we have implemented a rigorous approach to uncover the causal relationships among observed variables by employing a DAG learning¹⁷. Bayesian networks, typically represented as DAGs, are commonly learned using the constraint-based methods established upon conditional independence testing and search-and-score methods²⁶. Unfortunately, these traditional methods are computationally demanding due to the combinatorial nature that the number of DAGs grows exponentially with the number of observed variables.

In this paper, to address this issue, we innovatively approached the DAG learning by converting the traditional combinatorial optimization into a

continuous constrained optimization problem to avoid the combinatorial nature and circumvent the enormous computational demand^{27,28}. Next, input features with high relevance to the response variable were identified based on the MB search algorithm in accordance with the DAGs. An MB was defined as the union of the parents (nodes connected above), children (nodes connected below), and other parents of those children¹⁸. Please see texts in lines 605–623 of the revised manuscript for more details.

Results of DAG learning and MB based feature screening are quoted from the revised supplementary information as follows:

Supplementary Figure 15. The causal relationship between each input feature and the total generation quantity of HW. It is calculated based on the 10 industrial sectors combined data set. The figure shows each feature represented by an edge on the outermost circle, with causal relationships between features connected by indicator strips. The beginning and arrow end of each strip

indicate the cause and effect, respectively. The red strips correspond to the feature sets (the parents of total generation quantity of HW) that are highly correlated with the target variable (total generation quantity of HW) as obtained through the Markov blanket search algorithm.

Supplementary Figure 16. The causal relationship between each input feature and the total generation quantity of HW. It is calculated based on each sector data set. (a)~(i) are industrial sectors of organic chemical materials manufacture (OCM), chemical pesticides manufacture (CPM), specialized chemical products manufacture (SCP), steel rolling and processing (SRP), metal wire and rope manufacture (MWR), metal surface treatment (MST), electronic circuits manufacture (ECM), biomass energy generation (BEG), and electricity generation using other sources (EGU), respectively. The figure shows each feature represented by an edge on the outermost circle, with causal

relationships between features connected by indicator strips. The beginning and arrow end of each strip indicate the cause and effect, respectively. The red strips correspond to the feature sets (the parents of total generation quantity of HW) that are highly correlated with the target variable (total generation quantity of HW) as obtained through the Markov blanket search algorithm.

Supplementary Figure 17. The causal relationship between each input feature and the generation quantity of MHW. It is calculated based on the 10 industrial sectors combined data set. The figure shows each feature represented by an edge on the outermost circle, with causal relationships between features connected by indicator strips. The beginning and arrow end of each strip indicate the cause and effect, respectively. The red, green, and blue strips correspond to highly correlated feature sets with generation quantity of MHW computed based on the Markov blanket search algorithm. Specifically, red strips correspond to the parent of generation quantity of MHW, green strips

refer to the child of MHW, and blue strips corresponds to other parents of the child of generation quantity of MHW.

Supplementary Figure 18. The causal relationship between each input feature and the generation quantity of MHW. It is calculated based on each sector data set. (a)–(d) are industrial sectors of steel rolling and processing (SRP), metal wire and rope manufacture (MWR), metal surface treatment (MST), electronic circuits manufacture (ECM). The figure shows each feature represented by an edge on the outermost circle, with causal relationships between features connected by indicator strips. The beginning and arrow end of each strip indicate the cause and effect, respectively. The red strips correspond to the feature sets (the parents of generation quantity of MHW) that are highly correlated with the target variable (total generation quantity of MHW) as obtained through the Markov blanket search algorithm.

In addition to the procedure of DAG learning, features selected based on the MB search algorithm were further adjusted according to features' importance and correlation because it was found that relying solely on MB-based features

sometimes led to a decrease in predictive performance when comparing with the baseline models utilizing all the features as input. This reduction might be because the MB learning algorithm might fail to identify some critical features during feature selection tasks caused by the strict assumption of data distribution, variable types, or correctness of criteria, during causal discovery, which thereby limited the faithfulness of DAG learning results^{19,20}. Therefore, to overcome this limitation, features with high importance on model prediction but not being identified by MB search algorithm were added as relevant features. Meanwhile, variables presented tight correlation with the newly added features but lower importance were screened out. Please see text changes in lines 474–502 and 624–647 of the revised manuscript.

After feature screening, the input features for the combined and sector-independent models, along with the corresponding model performances of R^2 and RMSE, are quoted from the revised supplementary information as follows:

Supplementary Table 13. Performances of models to predict total generation quantity of HW under different feature selection methods.

Models	R²	RMSE	Final Input Features
Combined	0.87	255.14	manufacturing processes (including: metal heat treatment, circuit board processing, rectification, other separations, chemical reaction, incineration, equipment maintenance, wastewater treatment, waste gas treatment), industrial sector, firm scale, COD, pH, wastewater discharge amount, Fe, Cr
OCM	0.77	230.41	manufacturing processes (including: rectification, other separations, chemical reaction, incineration, equipment maintenance, wastewater treatment, waste gas treatment), firm scale, pH, P, NH₃-N, wastewater discharge amount
CPM	0.74	386.71	manufacturing processes (including: rectification, other separations, chemical reaction, incineration, equipment maintenance, wastewater treatment, waste gas treatment), firm scale, pH, COD, NH₃-N
SCP	0.91	264.71	manufacturing processes (including: metal heat treatment, metal surface processing, rectification, other separations, chemical reaction, incineration, equipment maintenance, wastewater treatment, waste gas treatment), firm scale pH, P, COD, NH₃-N

Models	R²	RMSE	Final Input Features
SRP	0.92	423.39	manufacturing processes (including: machining process, metal heat treatment, metal surface processing, equipment maintenance, wastewater treatment, waste gas treatment), firm scale, pH, P
MWR	0.95	126.59	manufacturing processes (including: machining process, metal heat treatment, metal surface processing, circuit board processing, equipment maintenance, waste gas treatment), firm scale, pH, P, N, COD
MST	0.88	27.52	manufacturing processes (including: metal heat treatment, metal surface processing, circuit board processing, equipment maintenance, wastewater treatment, waste gas treatment), firm scale, pH, COD, P, wastewater discharge amount, Ni, Cr^{VI}
ECM	0.78	479.39	manufacturing processes (including: machining process, metal heat treatment, metal surface processing, circuit board processing, other separations, incineration, equipment maintenance, waste gas treatment), firm scale, pH, P, NH₃-N, wastewater discharge amount, Cu
BEG	0.87	337.58	manufacturing processes (including: metal heat treatment, metal surface processing, other separations, chemical reaction, equipment maintenance, waste gas treatment), firm scale, pH, P, COD, NH₃-N
EGU	0.95	149.98	manufacturing processes (including: metal heat treatment, metal surface processing, circuit board processing, other separations, incineration, equipment maintenance, waste gas treatment), firm scale, NH₃-N, pH, P, wastewater discharge amount

Supplementary Table 14. Performances of models to predict generation quantity of MHW under different feature selection methods

Models	R²	RMSE	Final Input Features
Combined	0.80	45.98	manufacturing processes (including: machining process, metal heat treatment, metal surface processing, circuit board processing, rectification, other separations, chemical reaction, incineration, equipment maintenance, wastewater treatment, waste gas treatment), industrial sector, firm scale, pH, wastewater discharge amount, Cr, Cr^{VI}, Cu, Fe, Ni

Models	R²	RMSE	Final Input Features
SRP	0.80	32.92	manufacturing processes (including: machining process, metal heat treatment, metal surface processing, equipment maintenance, wastewater treatment, waste gas treatment), firm scale, wastewater discharge amount, pH, Cr, Cr^{VI}, Ni
MWR	0.84	20.91	manufacturing processes (including: machining process, metal heat treatment, metal surface processing, circuit board processing, wastewater treatment, waste gas treatment), firm scale, wastewater discharge amount, P, pH, NH₃-N, Zn
MST	0.68	30.86	manufacturing processes (including: machining process, metal heat treatment, metal surface processing, circuit board processing, equipment maintenance, wastewater treatment, waste gas treatment), firm scale, wastewater discharge amount, pH, P, Ni, Fe, Cu, Cr, Cr^{VI}
ECM	0.88	122.47	manufacturing processes (including: machining process, metal heat treatment, metal surface processing, circuit board processing, other separations, incineration, equipment maintenance, wastewater treatment, waste gas treatment), firm scale, COD, P, pH, N, Cu

5. **Line 137** The allocation of water and solids in the species association should be a phase allocation process at each stage of the process and should therefore be tailored to the elemental composition of hazardous wastes. However, since the types of hazardous wastes involved are not clearly defined, it is challenging to determine if the 12 water variables can fully describe all hazardous wastes.

Authors' response:

Suggestion taken. In our pursuit of developing a generic framework capable of predicting various categories of HW, we extensively investigated HW categories and wastewater pollutants based on China Environmental Statistics Database, national/industrial regulations, and literature review for 50 typical industrial sectors which contributed to 94.2% of the total HW generation quantity in China in 2015 (See Supplementary Table 4)

According to the thorough investigation, in the generic framework, we extended the 12 wastewater variables to 24, including wastewater discharge amount, chemical oxygen demand, pH, ammonia nitrogen, total nitrogen, total

phosphorus, suspended substance, petroleum hydrocarbon pollutants, biochemical oxygen demand, volatile phenol, total organic carbon, sulphate, fluoride, cyanide, lead, arsenic, cadmium, mercury, iron, total chromium, hexavalent chromium, copper, zinc, and nickel. Please see text changes in lines 140–156 of the revised manuscript.

Furthermore, to evaluate whether these 24 wastewater indicators could well describe HW generation, we have made a detailed literature review (including 71 papers) on the relationship between different wastewater contaminants and 46 types of HW in the list issued by the Ministry of Ecology and Environment of China²⁵. Results showed that 43 types of HW present tight affinities with some of the 24 wastewater variables (See Supplementary Table 1).

However, we acknowledge that for some types of HW, involving more representative contaminants might further improve the model's predictive performance. For instance, the inclusion of data on PAH discharge amount in wastewater can help to predict residues of rectifying and distillation (referring to code HW11 in the HW list issued by the Ministry of Ecology and Environment of China²⁵) from coal coking²⁹. Therefore, in the section of Limitation and future research, we stressed that under the generic framework of combining static enterprise characteristics and real-time manufacturing activities to predict HW generation, specific variables involved were flexible. If possible, more crucial variables could be included to further improve the model performance. The statement of this perspective can be found in lines 582–589 of the revised manuscript.

6. **Line 182** The size of RSME should refer to the mean value, which should be labeled or indicated in Figure 3.

Authors' response:

Thanks for your comments. To help readers have a better understanding of the evaluation metrics, we have added the calculation formulation of the model performance metrics, including RMSE, in the revised supplementary information. Please see texts in lines 193–226 of the revised supplementary information.

The RMSE in the Figure 3 referred to the comparison between prediction value and true value of the testing dataset⁵. It should be clarified that this RSME was not the mean value because there was only one testing dataset, ensuring an accurate comparison of predictive performance across different models, including those before and after data balance, combined and sector-independent models. However, a 10-fold cross-validation was conducted during model training to determine optimum hyperparameters (average performance on the 10 validation sets)⁵.

7. **Line 201** Does the performance optimization here compare to other interpolation methods?

Authors' response:

Suggestion taken. To demonstrate the superiority of the data balance method of Synthetic Minority Over-Sampling Technique for Regression with Gaussian Noise (SMOIGN), we have compared this method with two other commonly adopted over-sampling techniques of Synthetic Minority Over-Sampling Technique for Regression (SMOTER) and random over-sampling. SMOTER generates new synthetic examples from the rare cases through interpolating between two rare cases (one is a seed case and the other is randomly selected from the k-nearest neighbors of the seed) and the new synthetic sample is determined as a weighted average of the target variable values of the two rare cases³⁰. All rare cases are used in turn as seed examples. The major difference between the SMOTER and SMOIGN method is that SMOIGN introduced Gaussian Noise when the two rare cases are 'more distant'. Random over-sampling involves randomly sampling observations from rare cases and combining them with original rare cases to form the new dataset³¹.

Comparison results showed that the data balance method of SMOIGN outperformed other techniques (See Supplementary Table 23). Please see texts in lines 136–160 of the revised supplementary information and Supplementary Table 23 for more details. For the editor and reviewers' convenience, we quote the Supplementary Table 23 about comparison of different data balance approaches on model performances as follows:

Supplementary Table 23. Comparison of different data balance approaches on model performances

Data balance approaches		R^2	RMSE	MAD	MAE	MAP E	MSE	SSE
Total generation quantity of HW	Non data balance	0.80	314.83	49.24	143.24	52.90	99117.93	3.2E+08
	Random over-sampling	0.79	320.15	56.87	125.63	87.35	102496.02	3.3E+08
	SMOTER	0.85	265.16	61.41	124.07	64.40	70311.06	2.3E+08
	SMOBN	0.87	255.03	62.37	122.57	54.97	65039.69	2.1E+08
Generation quantity of MHW	Non data balance	0.80	54.12	1.80	12.30	-	2929.27	9.5E+06
	Random over-sampling	0.78	47.98	2.35	14.60	-	2302.55	7.5E+06
	SMOTER	0.86	37.89	1.46	11.68	-	1435.52	4.7E+06
	SMOBN	0.90	36.00	1.03	11.83	-	1295.78	4.2E+06

8. **Line 211** Section 3.1 and Section 3.2 can be combined because the second half of 3.1 and 3.2 deals with the handling of model boundary values.

Authors’ response:

Suggestion taken. We have restructured the manuscript by combining the contents of the previous sections 3.1 and 3.2 into a new consolidated section of ‘Combined model to predict generation quantity of hazardous waste at the firm-level’. This revised section provided a comprehensive analysis of the performance of both a combined model, predicting the total generation quantity of HW across 10 diverse industrial sectors, and a combined ensemble model dedicated to predicting generation quantity of metal surface treatment hazardous waste (MHW), a specific category of hazardous waste, across the same 10 different industrial sectors.

The reorganized section 3.1, mainly focusing on the combined model across 10 different sectors, consisted of two parts. The first part explored the selection of the best algorithm for building the model to predict the total generation quantity of HW and investigates the impact of data balancing on model performance. The second part presents the performance of the ensemble model, incorporating classification and regression models to predict the generation quantity of MHW, and compares the ensemble model with a direct regression model, highlighting the superiority of the ensemble model.

The detailed information about the new section of ‘Combined model to predict generation quantity of hazardous waste at the firm-level’ could be found in lines 195–264 of the revised manuscript.

9. **Line 254** For the study of Section 3.3 Sector-independent model development, it is suggested to use the analysis of three cases to compare with others, emphasizing the importance and universal practicability of the method.

Authors' response:

Thanks for your comments. In this section, we compared the predictive performance of the combined model with 9 sector-independent models for predicting the generation quantity of HW. The aim was to assess whether the development of sector-independent models could improve overall performance. Additionally, for the model dedicated to predicting the generation quantity of MHW, the combined model was compared with 4 sector-independent models.

Results showed that the performances of independent models were superior to the combined model, particularly for the sector with unsatisfactory prediction accuracy from the combined model. For example, the R^2 values of the independent models for the sector of metal surface treatment (MST) ($R^2=0.88$) and Biomass Energy Generation (BEG) ($R^2=0.87$) industry were much higher than the combined model ($R^2 = 0.82$ for MST and $R^2 = 0.73$ for BEG). This discrepancy was because combined model provided a global optimal solution rather than a local optimal solution³². Given the high diversity in HW generation patterns among sectors, sector-independent models were found to better simulate the patterns for each specific sector. Therefore, when there is sufficient data for each sector, it is recommended to build sector-independent models. Please see texts in lines 265–303 of the revised manuscript.

10. **Line 298** In the discussion of Section 3.4 Influence of input features on predicting hazardous waste generation, there is a lack of support in terms of material nature, production technology level, management level and manpower level.

Authors' response:

Thanks for your comments. In this section, the results were derived from SHAP analysis which quantifies the importance of input features on the response variable based on the marginal effects of the input feature. The factor of manpower level was represented by the feature of firm scale in this study because we employed the number of staff, which has been widely used as an indicator of manufacturing activities, to determine the firm scale and it was

classified into five categories (Supplementary Table 2) according to Chinese Criteria for the Division of Large, Medium and Micro Enterprises³³. SHAP analysis revealed that for the firm scale, the samples with low feature values were mainly on the left side, while the points with high feature values were mainly on the right side, suggesting the positive relationship between the firm scale and HW generation. It is consistent with the common sense that larger enterprises tend to generate more HW. Please see texts in lines 322–328 of the revised manuscript for more details.

While factors like material nature, production technology, and management level were not direct input features of the model, their influence on predicting HW generation cannot be directly quantified. Even so, we admit that these factors serve as underlying driving forces in the relationship between wastewater and HW generation. Therefore, additional discussion has been included regarding the potential impact of production technology and management level on the partition of contaminants between the liquid and solid phases. The discussion emphasized how these factors contribute to the high heterogeneity in the contaminant partition ratio among enterprises. It is also noted that this heterogenous relationship can be simulated through machine learning algorithms when wastewater and HW generation data are combined with data about a firm's static characteristics. The newly added discussion about the potential impact of production technology and management level on the partition of contaminants between the liquid and solid phases could be found in lines 329–349 of the revised manuscript.

11. **Line 308** Before determining the importance of features, please make sure that the 18 variables of the industrial sector, the firm scale, and manufacturing processes are independent.

Authors' response:

Suggestion taken. To ensure that input features are both useful and independent, we have added a new procedure of feature engineering, incorporating the causal discovery, importance ranking, and correlation analysis, in the model framework. To be specific, initially, the causal relationship was analyzed via DAG learning, where we derived a set of cause-and-effect relationships from the observational data, and the minimal set of features with high relevance to the response variable was obtained based on the MB search algorithm. Then,

features with high importance on model prediction but not being identified as relevant features were added. Meanwhile, variables presented tight correlation with the newly added features but lower importance were screened out to make sure that input variables were independent.

For more details about the feature engineering, please see text changes in lines 591–647 of the revised manuscript.

12. **Line 380** Hopefully, there is a summary group diagram to express the results.

Authors' response:

Suggestion taken. A summary diagram has been prepared to visually illustrate the generalized relationship between HW generation and input features for a total of 20 models, including a combined model and 9 sector-independent models to predict total generation quantity of HW, 5 classification models and 5 regression models from the combined and independent ensemble model to predict generation quantity of MHW. This summary diagram provided a comprehensive overview of the relationships captured by the different models.

For the editor and reviewers' convenience, we quote the newly revised Figure 6 as follows:

Figure 6. Importance of input features for 20 models, including a combined model and 9 sector-independent models to predict total generation quantity of HW, 5 classification models and 5 regression models from the combined and independent ensemble model to predict MHW generation quantity. (a) The average relative importance, indicated by the average MAS value of variables in 4 groups of the firm scale, manufacturing processes, the wastewater routine monitoring indicators, and metal emission in wastewater. of all the variables in one group (b) Ranking of input feature importance for 20 models. The number inside each cell reflected the ranking of the variable. Variables ranked in the top 5 are marked with * to denote their crucial importance in model prediction, while variables ranked between 6th and 10th are marked with * to indicate their relatively significant impact.**

This figure consisted of two parts. The first part, focusing on variable groups, presented the average relative importance of variables within 4 groups of the firm scale, manufacturing processes, the wastewater routine monitoring indicators, and metal emission in wastewater. Then in the second part, the rank of each input feature's importance was displayed for 20 models.

Generally, across 20 models, the feature groups of firm scale and routine monitoring indicators for wastewater, including wastewater discharge amount and emission of common pollutants (COD, NH₃-N, and N), were found to have high importance. Nonetheless, the importance of feature groups of manufacturing processes and metal emission in wastewater were generally lower and sector-dependent.

13. **Line 414** In the writing of Policy Implication, it is necessary to strengthen the expression of actionable and enforceable ideas and highlight the technical significance of innovation.

Authors' response:

Suggestion taken. To further evaluate whether this newly developed model framework is actionable and enforceable, we have organized a new section of Discussion to clearly delineate the generalizability of this model framework, sensitivity and uncertainty, and cost-effectiveness of this approach in practical applications.

Firstly, is there a correlation between start-up/retrofit time and the model? In other words, does the advanced nature of technology affect the proportion of solid liquid pollution?

Authors' response:

In the section of sensitivity analysis, to investigate the robustness of the model, we estimated the temporal extrapolation of the model during application, referring to how the performance of the model changed with prediction times, because the model was commonly built using the historic data but used to predict current HW generation in management practice. Specifically, the model, trained using data from Jan. 2020 to Dec. 2021, was employed to predict HW generation for each month in 2022.

The results indicated consistent and satisfactory performance of the combined model, with an R^2 around 0.76 when predicting the total generation quantity of HW for the next three months (Supplementary Figure 10). However, as the extrapolation timeframe increased, the performance gradually decreased to R^2 of 0.4-0.5. This decline might be attributable to a shift in the HW generation pattern not learned by the model using historical data. A similar trend can be found for the combined model to predict generation quantity of MHW. Therefore, we recommended periodic retrofitting and retraining of the model every three months to ensure accuracy during application. Please see text changes in lines 503–518 of the revised manuscript.

Secondly, can regional water conservation policies impact the results of the model? Since the study area encompasses multiple watersheds, water prices and availability are not constant.

Authors’ response:

It is possible that the regional water conservation policies will impact the enterprise’s water consumption level during manufacturing processes, which thereby change contaminant partition between liquid and solid phases. Aiming at this question, firstly, we adopted the ratio of wastewater discharge amount to HW generation quantity (WTH) to indicate the contaminant partition between liquid and solid phases and analyzed the Pearson’s correlation between enterprise’s water consumption level and the ratio of WTH. This analysis, based on data from 9,202 enterprises across China in 2015, aimed to determine whether variations in water consumption levels would influence contaminant partition between liquid and solid phases.

The findings, presented in Table R2, indicated that there was no significant correlation between the enterprise's water consumption level and the WTH ratio for most industrial sectors. It indicated that enterprise’s water consumption level did not have affinity with contaminant partition between liquid and solid phases.

Table R2. Pearson's correlation between enterprise’s water consumption level and the ratio of WTH for industrial sectors which had >30 observations.

Industrial sector	Pearson’s correlation	p-value
Mining and dressing of asbestos and mica	-	-

Industrial sector	Pearson's correlation	p-value
Machine-made paper and cardboard manufacturing	0.01	0.91
Lead and zinc metallurgy	-0.01	0.86
Gold mining	-0.10	0.67
Pigment manufacturing	0.05	0.61
Organic chemical materials manufacturing	-0.01	0.83
Electricity generation using other sources	-	-
Oil extraction	0.02	0.88
Gold smelting	0.09	0.78
Crude oil processing and petroleum product manufacturing	-0.04	0.57
Electronic circuits manufacturing	-0.02	0.76
Acrylic fiber manufacturing	-	-
Inorganic salt manufacturing	-0.06	0.51
Steel rolling and processing	-0.05	0.32
Chemical drug API manufacturing	-	-
Manufacturing of other basic chemical raw materials	-0.02	0.74
Thermal power generation	-0.05	0.64
Phosphate fertilizer manufacturing	-0.09	0.65
Manufacturing of wood and bamboo pulp	-	-
Metal surface treatment and heat treatment	0.04	0.06
Copper smelting	-0.05	0.71
Steelmaking	0.01	0.95
Coking	-0.02	0.73
Primary form plastic and synthetic resin manufacturing	-0.02	0.79
Iron making	-	-
Aluminum smelting	-0.07	0.64
Metal scrap and scrap processing	-	-
Vinylon fiber manufacturing	-	-
Automobile manufacturing	0.00	0.98
Other battery manufacturing	-0.02	0.85
Information chemical manufacturing	-0.06	0.69
Synthetic fiber mono (polymer) body manufacturing	-0.05	0.75
Dye manufacturing	-0.03	0.76
Manufacturing of architectural ceramic products	0.07	0.60
Manufacturing of optoelectronic devices and other electronic devices	-0.05	0.71
Auto parts and accessories manufacturing	-0.04	0.29
Chemical pesticides Manufacturing	-0.02	0.74
Tin smelting	-	-
Electronic components and component manufacturing	-0.02	0.79
Metal wire and rope manufacturing	0.19	0.00
Paint manufacturing	-0.01	0.85
Manufacturing of chemical reagents and auxiliaries	-0.01	0.84

Industrial sector	Pearson's correlation	p-value
Integrated circuit manufacturing	-	-
Fabrication of metal structures	-0.01	0.84
Copper rolling	-0.02	0.81
Specialized chemical products manufacturing	0.08	0.36
Inorganic alkali manufacturing	-0.08	0.59
Nickel and cobalt smelting	-	-
Other electronic equipment manufacturing	-0.08	0.52
Ferroalloy smelting	-0.08	0.60

Even so, the WTH ratio did exist difference among various regions. Take the sector of metal surface treatment (MST) for example. We used Kruskal-Wallis Test method to determine whether there were significant differences among various regions in the ratio of WTH²². It was obvious that even for the same sector, some regions displayed significant difference in the contaminant partition ratio between liquid and solid phase ($p < 2.2e^{-16}$). Further analysis was conducted using the Dunn's multiple comparison test (Table R3) to determine the level of significance of differences between each regions³⁴.

Table R3. Significance of differences among various regions in the ratio of WTH for the sector of metal surface treatment (MST) based on Dunn's multiple comparison test.

	Anhui	Fujian	Guangdong	Hebei	Jiangsu	Liaoning	Shandong	Shanghai	Sichuan	Tianjin
Fujian	0.31	-	-	-	-	-	-	-	-	-
Guangdong	0.73	0.00	-	-	-	-	-	-	-	-
Hebei	0.89	0.17	0.24	-	-	-	-	-	-	-
Jiangsu	0.74	0.00	0.94	0.30	-	-	-	-	-	-
Liaoning	0.84	0.09	0.94	0.65	0.96	-	-	-	-	-
Shandong	0.00	0.01	0.00	0.00	0.00	0.00	-	-	-	-
Shanghai	0.08	0.00	0.00	0.00	0.00	0.14	0.00	-	-	-
Sichuan	0.90	0.12	0.83	0.73	0.85	0.91	0.00	0.08	-	-
Tianjin	0.53	0.01	0.65	0.27	0.65	0.71	0.00	0.52	0.62	-
Zhejiang	0.00	0.00	0.00	0.00	0.00	0.00	0.00	0.00	0.00	0.00

Note: <0.05 means there is significant difference.

Then, to demonstrate how our generic model framework can be used to predict the HW generation in other regions with significantly different ratios of WTH, two cases were crafted to illustrate the model's applicability. Specifically, the

first application focused on the developing a model for the MST sector in Shandong province which had lower ratio of WTH than the studied region of Jiangsu. The sector of MST was chosen because it had sufficient sample size to build models for most provinces. The second focuses on model development in Zhejiang province which had significantly higher WTH than the studied region of Jiangsu.

For these two cases developed for regions with different WTH ratios, models presented commendable predictive performances (Shandong province: $R^2=0.69$, RMSE=29.38; Zhejiang province: $R^2=0.72$, RMSE=721.15). This suggested that our model framework adeptly captured and simulated the heterogeneous relationship between wastewater and HW generation, even when it varied across regions. In addition, when applying the model developed for Zhejiang to predict HW generation in Shanghai, a region in the same cluster as Zhejiang based on WTH ratio, the prediction performance fell below expectations ($R^2=0.32$) This implied that the model must be trained using the localized data derived from the application region. Please see text changes in lines 429–473 and Table 1 of the revised manuscript.

Comments by reviewer: 3

Overview:

In this paper, the authors utilized real-time wastewater data and static data to conduct firm-level estimation of hazardous waste generation. The execution of this study has been comprehensive, yielding some useful conclusions for waste management.

Authors' response:

We would like to thank the reviewer for the positive feedback on the robustness of the model design.

However, the innovativeness of this study is somewhat lacking, particularly in its methodology, where the proposed methods are the combinations of common traditional machine learning algorithms.

Authors' response:

Thanks for your comments. As we stated in the Introduction, the significance and novelty of this work mainly refer to the following three aspects: (a) scientific importance of comprehensively discovering and simulating the relationship between wastewater and hazardous waste (HW) generation, which provides a new perspective to address the intractable issue of HW source estimation at the firm-level; (b) the superiority of the methodology that put forward a generic framework to capture the heterogenous relationship between wastewater and HW generation through involving representative variables covering most of industrial sectors, adopting the advanced approach of directed acyclic graph (DAG) learning to uncover the casual relationship in feature engineering, combining data-balance preprocessing with machine learning algorithms to deal with imperfect long-tail data distribution, and conducting quantile regression forest to quantify the model uncertainty; (c) big progress in bestowing real environmental practice because this study offered a novel and cost-effective tool to conduct firm-level estimations of HW generation from numerous enterprises with great heterogeneities, an issue that has remained unresolved in its entirety.

Moreover, the outcomes yielded by the proposed methods are less than satisfactory, with R^2 values of about 0.8, which implies that the proposed methods lack practicality.

Authors' response:

Thanks for your comments. In this study, the case application to 1024 enterprises across 10 sectors in Jiangsu, China, achieved an R^2 value of 0.87 in predicting HW generation and we think that this performance is commendable, especially considering the inherent imperfections, such as long-tail distribution, in real-world data. It's crucial to note that our model outperformed traditional methods³⁵⁻³⁸ with comparable or even higher R^2 values, particularly when considering the increased temporal and spatial granular resolution achieved in our firm-level predictions. Many existing models with higher R^2 values often operate at coarser resolutions, such as the national or yearly level, sacrificing granularity for performance^{4,39-41}.

Secondly, to further demonstrate the reliability of the model during practical application, we have added uncertainty analysis in the revised manuscript. The results revealed that, for the combined models to predict generation quantity of HW, 95.85% testing samples fell into the 95% confidence intervals of prediction, indicating that the prediction value was reliability for management practice. Furthermore, we believe that, with continued application and enhancements in data quality, there is potential to further improve the predictive performance of our data-driven approach.

This work may be interesting for in hazardous waste management, but it may be difficult to generate interest in the broader research fields. Some other detailed comments are listed below for the authors' reference

Authors' response:

Thanks for your comments. This study has put forward a data-driven approach leveraging cross-domain big data, mainly involving enterprises' static characteristics and real-time activities, to track HW generation, a challenging task due to its high variability across diverse enterprises and the difficulty in automatically monitoring. Even though the studied case focused on HW, the proposed methodology holds broader applicability and can be extended to the prediction of other complex enterprises' behaviors, particularly those characterized by high uncertainty and being difficult to be tracked directly using Internet of Things sensors. For instance, the static characteristics and real-time compliance records of enterprises could be utilized to build machine learning models for predicting enterprise' environmental violations in the future, such as

private discharges after shutdowns, unauthorized changes in production processes, and concealment of pollutant types⁴². Furthermore, the static characteristics and real-time manufacturing activities of enterprises also enables the construction of machine learning models to predict firms' carbon emissions, addressing the lack of carbon reporting data for certain firms^{43,44}. Therefore, we argue that that the insights derived from this study extend beyond the realm of HW management and will capture the interest of readers in the broader field of environmental management.

Overall, in the case of Reviewer 3, whose recommendation relates more to the innovativeness, practicability, and significance of the model framework, we hope that we are able to satisfactorily address this concern in our earlier comments and the subsequent revisions we have made.

Below are point-to-point responses in which we explicitly refer to the relevant lines in the revised manuscript/supplementary information to clarify the genuine. The comments appear in black, our responses appear in indent and blue font, and the texts quoted from the revised manuscript/supplementary information appear in italics and green font.

Specific comments:

1. The author should engage in a more comprehensive discussion regarding the proposed scheme. From an enterprise perspective, does this scheme bear the potential to augment the company's expenditures? From a managerial standpoint, can this scheme yield dependable outcomes? And how cost-effective is the proposed scheme compared to other management measures?

Authors' response:

Suggestion taken. Combining the suggestion from Reviewer #2, in the revised manuscript, we have organized a comprehensive Discussion section to delineate the generalizability of this model framework, sensitivity and uncertainty, and cost-effectiveness of this approach in practice.

Firstly, regarding the reliability of the model results, we have conducted an uncertainty analysis using the method of quantile regression forest. Ideally, during uncertainty quantification, a reliable machine learning model should yield a c% confidence interval that contains the true value for approximately c%

of the time⁴⁵. For example, if $c\% = 95\%$, we expect that approximately 95% of test samples have their true values fall into the respective 95% confidence intervals of prediction. The uncertainty analysis in this study showed that our models exhibited low uncertainty, with 95.85% and 92.99% of testing samples falling within the respective 95% confidence intervals for the combined models predicting the generation quantity of HW and metal surface treatment hazardous waste (MHW, a typical category of HW in the studied case). This implied the high reliability in the model prediction results and good generalization of models. Please see text changes in lines 519–540 of the revised manuscript.

Secondly, the cost-effectiveness of this scheme was compared with traditional management measurement of field surveys from the perspective of time and monetary cost. When implemented on a standard personal computer (Intel Core i5-1135G7, 2.4 GHz CPU, and 16 GB memory), our model development required approximately 20 minutes, and the retrofit of model will take 12 minutes every 3 months. However, traditional field surveys to determine each enterprise's HW generation were time-consuming, often exceeding one day per enterprise⁴⁶ especially when the investigator lacked expertise in the specific manufacturing processes.

Then, in terms of the expenditure, our approach minimizes additional expenditures for companies, as it utilizes available wastewater data monitored by IoT sensors, which are increasingly mandated by environmental regulations in many countries. For example, Vietnam's Environmental Protection Law requires enterprises to install automatic pollution monitoring equipment⁴⁷. Even in the absence of such mandates, regulatory authorities commonly require industrial enterprises to periodically monitor and report wastewater data.^{48,49} Utilizing this data, it is still feasible to predict the HW generation during the same period as the reported time of wastewater data. As regulations tighten, necessitating more frequent reporting and the installation of automatic sensors,⁵⁰ the associated costs can be outweighed by the benefits, including optimized system operation and energy cost savings. Please see text changes in lines 541–576 of the revised manuscript.

In addition to these issues, in the section of Discussion, combining Reviewer #2's concern about the regional study, we have organized a part of 'Generalizability of this model framework' to extensively discuss how our

generic model framework can be applied in other regions and industrial sectors. The results showed that the framework presented commendable predictive performances in other regions and sectors, provided that model training data is localized, and variables related to manufacturing processes are appropriately screened based on the specific studied sector. For more detailed information, please see text changes in lines 428–473 of the revised manuscript.

Also, we have employed a series of sensitivity analyses, including determining the optimal model retrofit time and evaluating the effect of feature screening principles on model performance. The results suggested periodic retrofitting and retraining of the model every three months during application because there was consistent and satisfactory performance when predicting the total generation quantity of HW for the next three months. Regarding feature engineering, considering the computation efficiency as well as the model performance, it was satisfactory to select features through integrating causal discovery, importance ranking, and correlation analysis. For more detailed information, please see text changes in lines 474–518 of the revised manuscript.

2. As the authors have mentioned, “In the environmental management, lacking the ground-truth knowledge is the most intractable challenge, ...”, how can one ascertain the veracity of the data declared by the enterprises? If the data declared by the enterprises is unreliable, the prediction results of the proposed methods will be even more unreliable.

Authors’ response:

Suggestion taken. Combining the suggestion from Reviewer #2, we have made a detailed description of the data source and data processing to ensure the reliability of our data for model development.

Firstly, to ensure the credibility of the HW generation data declared by enterprises, we meticulously eliminated data from enterprises with records of environmental violations and administrative penalties. These records of environmental enforcement were sourced officially from the Department of Ecology and Environment of Jiangsu Province, China. Please see text changes in lines 46–66 of the revised supplementary information for more details.

Then, after data aggregation, we implemented a robust outlier screening and removal procedure, incorporating six unsupervised machine learning algorithms: K Nearest Neighbors, Minimum Covariance Determinant, Clustering-Based Local Outlier Factor, Histogram-based Outlier Score, Local Outlier Factor, and Isolation Forest. To minimize bias in outlier detection, the results from these algorithms were aggregated using the Average of Maximum (AOM) strategy²⁴. Please see texts in lines 68–94 of the revised supplementary information for more details.

3. Depending on the accuracy of the model (R^2 values are about 0.8), can the methods address the waste management problems mentioned by the authors?

Authors' response:

Thanks for your comments. To further evaluate the reliability of model prediction results, we have added a detailed uncertainty analysis in the revised manuscript. The results revealed that for the combined models to predict generation quantity of HW and MHW, 95.85% and 92.99% testing samples fell into the respective 95% confidence intervals of prediction. This highlights the reliability of our model predictions for effective management practices. Please see text change in lines 519–540 of the revised manuscript.

Furthermore, current performance with R^2 values of about 0.8 surpassed that of many existing models for predicting HW generation^{3,11}. In addition, it is noteworthy that some models achieving higher R^2 values often operate at lower temporal and spatial granular resolutions, such as national or yearly levels^{4,39-41}. In our study, the slight decrease in predictive performance is a deliberate trade-off, strategically made to enhance the model's resolution to the firm-level. We believe that with continued application of this approach and improvements in data quality, it has potential to further improve the predictive performance of this data-driven approach.

4. It is suggested that the author incorporate repetitive experiments to elucidate the reliability of the predictive outcomes.

Authors' response:

Suggestion taken. The reliability of the model was demonstrated through a rigorous 5-fold cross-test that the whole dataset was randomly split into 5 testing

datasets with each accounting for 20% of the whole dataset and the remaining observations were regarded as the training dataset. The machine learning model was trained and then applied on each testing dataset to evaluate the robustness and stability of its performance. This iterative approach allowed us to systematically assess the model's performance across different subsets of the data.

The average performance and standard deviation (SD) of the 5-fold cross-validation for different prediction models are shown in Tables R4 and R5. The consistently stable average value and low SD in both R^2 and RMSE indicate that our models are highly robust. This comprehensive validation approach enhances our confidence in the reliability of the proposed methodology.

Table R4. The average performance and standard deviation (SD) of the 5-fold cross-validation for the models to predict total generation quantity of hazardous waste.

Models	R^2 _Average	R^2 _SD	RMSE_Average	RMSE_SD
The Combined model	0.84	0.05	250.99	79.77
OCM	0.89	0.13	149.19	64.94
CPM	0.83	0.11	301.63	95.66
SCP	0.88	0.09	364.16	109.19
Independent models for each sector				
SRP	0.91	0.05	368.28	179.12
MWR	0.88	0.12	123.50	79.03
MST	0.91	0.05	16.02	6.53
ECM	0.88	0.13	346.20	182.26
BEG	0.88	0.07	251.59	115.26
EGU	0.93	0.05	153.99	44.01

Table R5. The average performance and standard deviation (SD) of the 5-fold cross-validation for the models to predict total generation quantity of metal surface treatment hazardous waste.

Models	R^2 _Average	R^2 _SD	RMSE_Average	RMSE_SD
The Combined model	0.72	0.12	69.99	19.56
Independent models for each sector				
SRP	0.77	0.11	34.76	13.32
MWR	0.85	0.13	22.00	8.28
MST	0.74	0.08	20.13	5.89
ECM	0.77	0.16	188.35	54.91

5. What is the number of missing values being filled in? Does the number of filled values affect the reliability of the model outcomes?

Authors' response:

Suggestion taken. We have calculated the number of missing value and presented it in the supplementary information (Supplementary Table 21). It is essential to note that the missing value imputation was conducted only for the routine wastewater monitoring indicators of chemical oxygen demand (COD), pH, ammonia nitrogen (NH₃-N), total nitrogen (N), and total phosphorus (P) because it was hypothesized that these contaminants commonly existed in wastewater. However, regarding the 6 metal emission indicators (iron (Fe), total chromium (Cr), hexavalent chromium (Cr^{VI}), copper (Cu), zinc (Zn), and nickel (Ni)), any missing values were treated as an indication that the respective firm did not generate the corresponding metal contaminant during its manufacturing processes.

For the editor and reviewers' convenience, we quote the results of missing values proportions for routine monitoring indicators of wastewater in the revised supplementary information as follows:

Supplementary Table 21. Proportion of missing values for 6 routine monitoring indicators of wastewater in the dataset of the studied cases.

Industrial sector	Data counts	Wastewater discharge amount	COD	pH	NH₃-N	N	P
OCM	3205	0.00%	4.12%	26.46%	38.91%	87.30%	80.41%
CPM	1891	0.00%	1.16%	15.02%	18.30%	85.30%	55.00%
SCP	1176	0.00%	4.17%	24.15%	39.88%	97.87%	90.31%
STE	28	0.00%	3.57%	0.00%	3.57%	-	42.86%
SRP	1192	0.00%	61.66%	61.91%	82.05%	91.28%	91.28%
MWR	833	0.00%	50.30%	52.94%	91.60%	97.00%	95.08%
MST	6909	0.00%	49.63%	40.86%	70.47%	94.56%	83.92%
ECM	1531	0.00%	10.12%	49.38%	32.33%	95.36%	89.61%
BEG	248	0.00%	1.21%	41.13%	27.82%	88.71%	88.71%
EGU	160	0.00%	14.38%	20.00%	14.38%	95.63%	73.75%

Even though some variables exhibited high percentages of missing values, even exceeding 90%, these variables were commonly screened out during feature engineering because they commonly presented tight correlation with the feature of wastewater discharge amount, but lower importance than it. The detailed information about input features after feature selection for the combined and

sector-independent models could be found in Supplementary Table 13 & 14.

Meanwhile, we admit that missing values and accompanying loss of information might impact the model performance, but such challenge is common because data is always imperfect in the reality. To overcome this limitation, we compared different methods of missing value imputation based on contaminant emission intensity, industrial sector median emissions, KNNimpute, and MissForest algorithm to select the one being most appropriate for this model framework.

The approach of missing value imputation based on contaminant emission intensity was to fill the missing value via multiplying the contaminant emission intensity by the wastewater discharge amount. In detail, the contaminant emission intensity, referring to the contaminant emission per wastewater flow, was the median value at the firm-level. If the enterprise had no records of contaminant, the median value at the industrial sectors level was used. The method of missing value imputation based on industrial sector medians is to use the median value of the industrial sector to fill the missing values. KNNimpute is an algorithm to impute missing values based on the weighted value of k-nearest neighbor samples^{51,52}. MissForest algorithm predicted the missing value using the random forest algorithm⁵³.

Then, it has been demonstrated that the strategy of imputing missing value through multiplying the contaminant emission intensity by the wastewater discharge amount emerged as the most effective approach during our comparison (See Supplementary Table 22). Based on the missing value imputation, comparison between the prediction value and true value demonstrated that the model could well predict the HW generation with the performance of R^2 at 0.87. This comprehensive analysis underscores the importance of selecting an appropriate strategy for missing value imputation, thereby contributing to the overall reliability and robustness of our model. Please see texts in lines 105–135 and Supplementary Table 22 of the revised supplementary information for more details.

For the editor and reviewers' convenience, we quote the Supplementary Table 22 about comparison of different missing value imputation approaches on model performances as follows:

Supplementary Table 22. Comparison of different missing value imputation approaches on model performances

Models	missing value imputation approaches	R²	RMSE	MAD	MAE	MAPE	MSE	SSE
The combined model to predict the total generation quantity of HW	Based on the industrial sector median emissions	0.23	2573.49	76.91	268.38	66.25	6.6E+06	2.0E+10
	KNNimpute	0.31	2772.55	96.47	253.70	66.54	7.7E+06	2.3E+10
	MissForest	0.81	574.19	102.36	171.58	69.35	3.3E+05	1.0E+09
	Based on the contaminant emission intensity	0.87	255.03	62.37	122.57	54.97	65039.69	2.1E+08
	Based on the industrial sector median emissions	0.48	100.42	4.30	17.49	-	1.0E+04	3.1E+07
	KNNimpute	0.49	70.75	70.75	17.34	-	5.0E+03	1.5E+07
generation quantity of MHW	MissForest	0.59	2295.52	6.07	65.39	-	5.3E+06	1.6E+10
	Based on the contaminant emission intensity	0.90	36.00	1.03	11.83	-	1295.78	4.2E+06

6. The author should elucidate the rationale behind employing staff number as a representation of a firm scale, given the varying degrees of labor intensity across different industries.

Authors' response:

Suggestion taken. We have illustrated the rationale behind employing staff number as a representation of a firm scale in the revised manuscript. Regarding the determination of the firm scale, we referred to the ‘Chinese Criteria for the Division of Large, Medium and Micro Enterprises’ proposed by National Bureau of Statistics of China³³ and it classified the enterprise’s scale based on employee number and annual income. Since the annual income of each’s enterprise is always confidential, considering the data availability, we adopted only the employee number to indicate the firm scale. Please see text changes in lines 121–125 of the revised manuscript.

We acknowledge that the labor intensity varies across different industrial sectors, so we further enhanced the combined model by incorporating the variable indicating the industrial sector to which the firm belongs. This addition allows the model to effectively capture and learn the interactions between the variables

of firm scale and industrial sector. Then, the good predictive performance of the model and the high importance of the variable of firm scale implied that such classification is reasonable.

References

- 1 Öncel, Mehmet Salim *et al.* Hazardous wastes and waste generation factors for plastic products manufacturing industries in Turkey. *Sustainable Environment Research* **27**, 188-194 (2017). <https://doi.org/https://doi.org/10.1016/j.serj.2017.03.006>
- 2 Karahan, Özlem, Tasli, Ruya, Dulekgurgen, Ebru & Görgün, Erdem. Estimation of hazardous waste factors. *Desalination and Water Treatment* **26**, 79-86 (2011).
- 3 Liu, Shuangliu, Cheng, Liang, Chen, Peng, Xu, Shunqing, Gao, Jun & Lu, Jing. Prediction of Industrial Hazardous Waste Production Based on Different Models. *IOP Conference Series: Earth and Environmental Science* **384**, 012026 (2019). <https://doi.org/10.1088/1755-1315/384/1/012026>
- 4 Karpušenkaitė, Aistė, Denafas, Gintaras & Ruzgas, Tomas. Forecasting Hazardous Waste Generation Using Short Data Sets: Case Study of Lithuania. *Mokslas - Lietuvos Ateitis* **8**, 357-364 (2016).
- 5 Zhong, Shifa *et al.* Machine Learning: New Ideas and Tools in Environmental Science and Engineering. *Environmental Science & Technology* **55**, 12741-12754 (2021). <https://doi.org/10.1021/acs.est.1c01339>
- 6 Rout, Prangya R., Zhang, Tian C., Bhunia, Puspendu & Surampalli, Rao Y. Treatment technologies for emerging contaminants in wastewater treatment plants: A review. *Science of The Total Environment* **753**, 141990 (2021). <https://doi.org/https://doi.org/10.1016/j.scitotenv.2020.141990>
- 7 Petrie, Bruce, Barden, Ruth & Kasprzyk-Hordern, Barbara. A review on emerging contaminants in wastewaters and the environment: Current knowledge, understudied areas and recommendations for future monitoring. *Water Research* **72**, 3-27 (2015). <https://doi.org/https://doi.org/10.1016/j.watres.2014.08.053>
- 8 Ghaithan, Ahmed, Khan, Mohammed, Mohammed, Awsan & Hadidi, Laith. Impact of Industry 4.0 and Lean Manufacturing on the Sustainability Performance of Plastic and Petrochemical Organizations in Saudi Arabia. *Sustainability* **13**, 11252 (2021).
- 9 Garetti, Marco & Taisch, Marco. Sustainable manufacturing: trends and research challenges. *Production Planning & Control* **23**, 83-104 (2012). <https://doi.org/10.1080/09537287.2011.591619>
- 10 Khan, Waseem Ullah, Ahmed, Sirajuddin, Dhoble, Yogesh & Madhav, Sugghosh. A critical review of hazardous waste generation from textile industries and associated ecological impacts. *Journal of the Indian Chemical Society* **100**, 100829 (2023). <https://doi.org/https://doi.org/10.1016/j.jics.2022.100829>
- 11 Lin, Kunsen, Zhao, Youcai & Kuo, Jia-hong. Data-driven models applying in household hazardous waste: Amount prediction and classification in Shanghai. *Ecotoxicology and Environmental Safety* **263**, 115249 (2023). <https://doi.org/https://doi.org/10.1016/j.ecoenv.2023.115249>
- 12 Yang, Jing, Guo, Huanxiu, Liu, Beibei, Shi, Rui, Zhang, Bing & Ye, Weili. Environmental regulation and the Pollution Haven Hypothesis: Do environmental regulation measures matter? *Journal of Cleaner Production* **202**,

- 993-1000 (2018). <https://doi.org/https://doi.org/10.1016/j.jclepro.2018.08.144>
- 13 Province, Department of Ecology and Environment of Jiangsu. *Measures for the Administration of Automatic Monitoring of Pollution Sources of Jiangsu Province*. (2022).
- 14 Biau, Gérard. Analysis of a random forests model. *The Journal of Machine Learning Research* **13**, 1063–1095 (2012).
- 15 Duch, Włodzisław & Jankowski, Norbert. Transfer functions: hidden possibilities for better neural networks. *ESANN*, 81-94 (2001).
- 16 Sharma, Sagar, Sharma, Simone & Athaiya, Anidhya. Activation functions in neural networks. *Towards Data Sci* **6**, 310-316 (2017).
- 17 Scholkopf, Bernhard. Causality for Machine Learning. *Probabilistic and Causal Inference* (2019).
- 18 Pellet, Jean-Philippe & Elisseeff, André. Finding Latent Causes in Causal Networks: an Efficient Approach Based on Markov Blankets. *Neural Information Processing Systems* (2008).
- 19 Wu, Xingyu, Jiang, Bingbing, Wu, Tianhao & Chen, Huanhuan. Practical Markov boundary learning without strong assumptions. *Proceedings of the AAAI Conference on Artificial Intelligence* **37**, 10388-10398 (2023).
- 20 Margaritis, Dimitris. Toward provably correct feature selection in arbitrary domains. *Advances in neural information processing systems* **22** (2009).
- 21 Miranda, Lorena S., Deilami, Kaveh, Ayoko, Godwin A., Egodawatta, Prasanna & Goonetilleke, Ashantha. Influence of land use class and configuration on water-sediment partitioning of heavy metals. *Science of The Total Environment* **804**, 150116 (2022).
<https://doi.org/https://doi.org/10.1016/j.scitotenv.2021.150116>
- 22 Choi, Sol, Ekpe, Okon Dominic, Sim, Wonjin, Choo, Gyojin & Oh, Jeong-Eun. Exposure and Risk Assessment of Korean Firefighters to PBDEs and PAHs via Fire Vehicle Dust and Personal Protective Equipment. *Environmental Science & Technology* **57**, 520-530 (2023). <https://doi.org/10.1021/acs.est.2c06393>
- 23 Wang, Yazhu, Deng, Yawen, Duan, Xuejun, Zou, Hui & Wang, Lingqing. Spatial correlation and coupling between industrial enterprise agglomeration and water pollutant discharge. *Chemosphere* **341**, 139752 (2023).
<https://doi.org/https://doi.org/10.1016/j.chemosphere.2023.139752>
- 24 Aggarwal, Charu C. & Sathe, Saket. Theoretical Foundations and Algorithms for Outlier Ensembles. *SIGKDD Explor. Newsl.* **17**, 24–47 (2015).
<https://doi.org/10.1145/2830544.2830549>
- 25 Ministry of Ecology and Environment of the People's Republic of China. *Chinese National List of Hazardous Wastes*. (2021).
- 26 Castelletti, Federico & Consonni, Guido. Discovering Causal Structures in Bayesian Gaussian Directed Acyclic Graph Models. *Journal of the Royal Statistical Society Series A: Statistics in Society* **183**, 1727-1745 (2020).
<https://doi.org/10.1111/rssa.12550>
- 27 Zheng, Xun, Aragam, Bryon, Ravikumar, Pradeep & Xing, Eric P. DAGs with NO TEARS: continuous optimization for structure learning. *Proceedings of the*

- 32nd International Conference on Neural Information Processing Systems, 9492–9503 (2018).
- 28 Zhang, Xiaoge, Wang, Xiao-Lin, Fan, Fenglei, Cheung, Yiu-Ming & Bose, Indranil. Enhancing the Performance of Neural Networks Through Causal Discovery and Integration of Domain Knowledge. *ArXiv* **abs/2311.17303** (2023).
- 29 Kong, Qiaoping *et al.* Solubilization of polycyclic aromatic hydrocarbons (PAHs) with phenol in coking wastewater treatment system: Interaction and engineering significance. *Science of The Total Environment* **628-629**, 467-473 (2018). <https://doi.org/https://doi.org/10.1016/j.scitotenv.2018.02.077>
- 30 Torgo, Luís, Ribeiro, Rita P., Pfahringer, Bernhard & Branco, Paula. SMOTE for Regression. *16th Portuguese Conference on Artificial Intelligence (EPIA)* **8154**, 378-389 (2013).
- 31 Santoso, B, Wijayanto, H, Notodiputro, KA & Sartono, B. Synthetic over sampling methods for handling class imbalanced problems: A review. *IOP conference series: earth and environmental science* **58**, 012031 (2017).
- 32 Yang, Hongrui, Huang, Kuan, Zhang, Kai, Weng, Qin, Zhang, Huichun & Wang, Feier. Predicting Heavy Metal Adsorption on Soil with Machine Learning and Mapping Global Distribution of Soil Adsorption Capacities. *Environmental Science & Technology* **55**, 14316-14328 (2021). <https://doi.org/10.1021/acs.est.1c02479>
- 33 China, National Bureau of Statistics of. *Chinese Criteria for the Division of Large, Medium and Micro Enterprises*. (2011).
- 34 Castiglioni, Sara, Borsotti, Andrea, Senta, Ivan & Zuccato, Ettore. Wastewater Analysis to Monitor Spatial and Temporal Patterns of Use of Two Synthetic Recreational Drugs, Ketamine and Mephedrone, in Italy. *Environmental Science & Technology* **49**, 5563-5570 (2015). <https://doi.org/10.1021/es5060429>
- 35 Delavaux, Camille S. *et al.* Native diversity buffers against severity of non-native tree invasions. *Nature* **621**, 773-781 (2023). <https://doi.org/10.1038/s41586-023-06440-7>
- 36 Lee, Jaehyun *et al.* Soil organic carbon is a key determinant of CH₄ sink in global forest soils. *Nature Communications* **14**, 3110 (2023). <https://doi.org/10.1038/s41467-023-38905-8>
- 37 Zhan, Yu *et al.* Satellite-Based Estimates of Daily NO₂ Exposure in China Using Hybrid Random Forest and Spatiotemporal Kriging Model. *Environmental Science & Technology* **52**, 4180-4189 (2018). <https://doi.org/10.1021/acs.est.7b05669>
- 38 Ratledge, Nathan, Cadamuro, Gabe, de la Cuesta, Brandon, Stigler, Matthieu & Burke, Marshall. Using machine learning to assess the livelihood impact of electricity access. *Nature* **611**, 491-495 (2022). <https://doi.org/10.1038/s41586-022-05322-8>
- 39 Shi, Wenjie, Zhao, Youcai, Li, Zongsheng, Zhang, Wenxiao, Zhou, Tao & Lin, Kunsen. Transformer-based enhanced model for accurate prediction and

- comprehensive analysis of hazardous waste generation in Shanghai: Implications for sustainable waste management strategies. *Chemosphere* **338**, 139579 (2023).
<https://doi.org/https://doi.org/10.1016/j.chemosphere.2023.139579>
- 40 Adamović, Vladimir M., Antanasijević, Davor Z., Ristić, Mirjana Đ, Perić-Grujić, Aleksandra A. & Pocajt, Viktor V. An optimized artificial neural network model for the prediction of rate of hazardous chemical and healthcare waste generation at the national level. *Journal of Material Cycles and Waste Management* **20**, 1736-1750 (2018). <https://doi.org/10.1007/s10163-018-0741-6>
- 41 Ceylan, Zeynep, Bulkan, Serol & Eevli, Sermin. Prediction of medical waste generation using SVR, GM (1,1) and ARIMA models: a case study for megacity Istanbul. *Journal of Environmental Health Science and Engineering* **18**, 687-697 (2020). <https://doi.org/10.1007/s40201-020-00495-8>
- 42 Zhou, Qi, Qu, Shen, Wang, Qianzi, She, Yunlei, Yu, Yang & Bi, Jun. Sliding Window-Based Machine Learning for Environmental Inspection Resource Allocation. *Environmental Science & Technology* **57**, 16743-16754 (2023). <https://doi.org/10.1021/acs.est.3c05088>
- 43 Heurtebize, Thibaut, Chen, Frederic, Soupé, François & Carvalho, Raul Leote de. Corporate carbon footprint: A machine learning predictive model for unreported data. *Available at SSRN 4038436* (2022).
- 44 Nguyen, Quyen, Diaz-Rainey, Ivan & Kurupparachchi, Duminda. Predicting corporate carbon footprints for climate finance risk analyses: A machine learning approach. *Energy Economics* **95**, 105129 (2021). <https://doi.org/https://doi.org/10.1016/j.eneco.2021.105129>
- 45 Nemani, Venkat *et al.* Uncertainty quantification in machine learning for engineering design and health prognostics: A tutorial. *Mechanical Systems and Signal Processing* **205**, 110796 (2023).
- 46 Berzi, Lorenzo, Delogu, Massimo, Giorgetti, Alessandro & Pierini, Marco. On-field investigation and process modelling of End-of-Life Vehicles treatment in the context of Italian craft-type Authorized Treatment Facilities. *Waste Management* **33**, 892-906 (2013). <https://doi.org/10.1016/j.wasman.2012.12.004>
- 47 Vietnam. *Law on Environmental Protection.*, Vol. No. 72/2020/QH14(2020).
- 48 Agency, U.S. Environmental Protection. *National Pollutant Discharge Elimination System (NPDES)*. (1972).
- 49 European Commission, Directorate-General for Environment *Commission Implementing Decision (EU) 2022/2427 of 6 December 2022 establishing the best available techniques (BAT) conclusions, under Directive 2010/75/EU of the European Parliament and of the Council on industrial emissions, for common waste gas management and treatment systems in the chemical sector (notified under document C(2022) 8788)*. Vol. 2022/2427(2022).
- 50 Martínez, Ramón, Vela, Nuria, el Aatik, Abderrazak, Murray, Eoin, Roche, Patrick & Navarro, Juan M. On the Use of an IoT Integrated System for Water

- Quality Monitoring and Management in Wastewater Treatment Plants. *Water* **12**, 1096 (2020).
- 51 Troyanskaya, Olga *et al.* Missing value estimation methods for DNA microarrays. *Bioinformatics* **17**, 520-525 (2001).
<https://doi.org/10.1093/bioinformatics/17.6.520>
- 52 Silva, Jonathan de Andrade & Hruschka, Eduardo Raul. An experimental study on the use of nearest neighbor-based imputation algorithms for classification tasks. *Data & Knowledge Engineering* **84**, 47-58 (2013).
<https://doi.org/https://doi.org/10.1016/j.datak.2012.12.006>
- 53 Stekhoven, D. J. & Bühlmann, P. MissForest-non-parametric missing value imputation for mixed-type data. *Bioinformatics* **28**, 112-118 (2012).
<https://doi.org/10.1093/bioinformatics/btr597>

REVIEWER COMMENTS

Reviewer #2 (Remarks to the Author):

We acknowledge the authors' modifications to enhance feature engineering and the application of the discussion model, which will improve the paper's credibility and broaden its readership. It is suggested that the manuscript be published with minor revision.

There are no major flaws in the model evident in the article, but two details require verification.

(1) Has dimension reduction been implemented due to the extensive number of features to prevent information redundancy in section 3.1? This inquiry aims to mitigate the risk of overfitting.

(2) In the fitting outcomes across multiple provinces (section 4.2), it is imperative to ascertain if there are significant variations in feature importance and spatial heterogeneity.

Moreover, beyond the aforementioned model considerations, it may be beneficial to discuss the model's applicability scope and provide specific guidelines for its practical implementation in the final paragraph.

Reviewer #2 (Remarks on code availability):

The code cannot be read on the page.

Reviewer #3 (Remarks to the Author):

The authors have solved some of the problems and improved the quality of the manuscript, but the following major problems still need to be addressed before it can be accepted for publication.

(1) The authors have used five machine learning algorithms to achieve the prediction of HW, how about using currently popular deep learning algorithms? More comparisons and discussions about novel algorithms would make the exposition more complete and strength the importance of this work.

(2) As the authors show the results in lines 452-454, the model's R2 is 0.69 and 0.72 in the cases of Shandong Province and Zhejiang Province, respectively, while the value is 0.87 in the case of Jiangsu Province. What are the reasons for the differences in the model's effectiveness in different cases? How to solve this problem of poor generalization?

(3) As shown in Table Supplementary Table 21, some of the indicators are missing more than 80%, does the methods of missing value imputation cause information leakage, in other words, introducing information from the test dataset into the training dataset? Please kindly know that information leakage is one of the major reasons that cause the fictitious good prediction performance (Sustainable Cities and Society 94 (2023) 104541). The above related paper and other closely related ones are suggested to cited and discussed.

(4) The model framework shown in Figure 2 is incomplete, for example, the Data Preprocessing also includes missing value imputation. The completeness of the model framework is crucial for readers to understand and use the authors' ensemble model.

Itemized Replies to Review Comments

(Manuscript Number: NCOMMS-23-45555A)

Comments by reviewer #2

Overview:

We acknowledge the authors' modifications to enhance feature engineering and the application of the discussion model, which will improve the paper's credibility and broaden its readership. It is suggested that the manuscript be published with minor revision.

Authors' response:

We would like to thank the reviewer for the positive feedback on the modification and revision that have improved the paper's credibility and broadened its readership.

There are no major flaws in the model evident in the article, but two details require verification.

Authors' response:

Based on your comments, we have made point-by-point revision to further improve the quality of this paper. Below are point-to-point responses in which we explicitly refer to the relevant lines in the revised manuscript/supplementary information to clarify the genuine. The comments appear in black, our responses appear in indent and blue font, and the texts quoted from the revised manuscript/supplementary information appear in italics and green font.

Specific comments:

1. Has dimension reduction been implemented due to the extensive number of features to prevent information redundancy in section 3.1? This inquiry aims to mitigate the risk of overfitting.

Authors' response:

Thanks for your comment. With respect to the combined models for firm-level prediction of the hazardous waste (HW) generation quantity in Section 3.1, feature engineering has been implemented to reduce the number of input

features and prevent information redundancy, thereby mitigating the risk of model overfitting¹. We have provided detailed information on the input features selected after variable screening for each model in the Supplementary Table 10 and 12. Furthermore, to lower the risk of overfitting during model training, we implemented 10-fold cross-validation. This technique enables optimizing hyperparameters effectively, ensuring that the models generalize well to unseen data beyond the training set.

Here, we compared the model’s performance on the training data with its performance on the testing data to assess the risk of overfitting. It is found that from the perspective of both model prediction performance and overfitting risk, RF model performed best among the eight models. Specifically, the RF model demonstrated a slight difference in performance between the training and testing datasets, with the R^2 value on the training data being only approximately 10% higher than that on the testing data. This discrepancy is similar to the behavior commonly observed in machine learning models developed using real-world data across various fields, such as rainforest carbon emission estimation, agricultural ammonia emission estimation, and polymer membrane design (Table R3). Consequently, we conclude that the risk of overfitting associated with the RF model is low and falls within an acceptable range.

Table R1. Performance comparison of the combined model to predict the total generation quantity of HW on the training and testing data.

Models		Training dataset		Testing dataset	
		R^2	RMSE	R^2	RMSE
Machine Learning	GBDT	0.95	221.44	0.81	298.05
	XGB	0.96	189.86	0.81	296.17
	SVM	0.87	348.01	0.81	295.93
	KNN	0.96	198.82	0.80	290.70
	RF	0.95	219.49	0.87	247.40
Deep Learning	MLP	0.87	337.53	0.86	249.99
	Ensemble	0.89	314.62	0.87	241.09
	MLP	0.83	411.03	0.84	295.82
	TNN	0.83	411.03	0.84	295.82

Note that the test dataset used for evaluating the performance of all models is the same. Data balance is conducted only on the training dataset.

Table R2. Performance comparison of the combined ensemble model to predict the total generation quantity of MHW on the training and testing dataset.

Models		Training dataset		Testing dataset	
		R ²	RMSE	R ²	RMSE
Machine Learning	GBDT	0.72	102.07	0.48	116.34
	XGB	0.78	92.08	0.50	118.12
	SVM	0.71	99.47	0.54	87.35
	KNN	0.73	95.84	0.45	106.43
	RF	0.95	30.64	0.85	47.50
Deep Learning	MLP	0.92	90.06	0.67	68.63
	Ensemble	0.96	59.43	0.21	108.09
	MLP				
	TNN	0.89	95.65	0.56	96.32

Note that the test dataset used for evaluating the performance of all models is the same. Data balance is conducted only on the training dataset.

Table R3. Training and testing performance comparison of the machine learning/deep learning models developed in various application fields based on literature review

Author	Journal	Application field	Algorithms	Training performance (R ²)	Testing performance (R ²)
Xu et al. ²	Nature	Estimating Ammonia nitrogen emissions	RF	0.92	0.79
Guerra et al. ³	Nature	Predicting soil ecological hotspots	RF	0.86~0.91	0.18~0.63
Hsu et al. ⁴	Nature Communications	Assessing the benefits of CO ₂ emission reduction	XGB	0.99	0.87
Barnett et al. ⁵	Science Advances	Designing polymer membrane	GPR	0.98~0.99	0.80~0.90
Yu et al. ⁶	Science Advances	Assessing the impact of nanoparticles on immune burden	RF	>0.90	0.63~0.87
Wang et al. ⁷	Science Advances	Estimating carbon emissions from	ETR	0.89	0.75

Author	Journal	Application field	Algorithms	Training performance (R²)	Testing performance (R²)
		rainforests			

2. In the fitting outcomes across multiple provinces (section 4.2), it is imperative to ascertain if there are significant variations in feature importance and spatial heterogeneity.

Authors’ response:

Suggestion taken. In the section of 4.1 which we elucidated the adaptability of our generic model framework in predicting HW generation across diverse regions, we have enriched the analysis by including a feature importance assessment for each model. (Supplementary Fig. 12-14). Then, to gain deeper insights into the spatial heterogeneity of feature importance, we further extended the region of interest associated with the MST sector into another three regions (Guangdong, Hebei, and Fujian province) using the data from the same database of the China Environmental Statistics Database of 2015. Finally, we compared the performance of six models developed for the MST sector across the regions of Shandong, Zhejiang, Jiangsu, Guangdong, Hebei, and Fujian provinces. These regions were selected based on considerations of both data availability and their representativeness in capturing diverse contaminant partition relationships between the solid and liquid phases. The spatial distributions of these six regions are shown in Figure R1.

Figure R1. Studied regions for six models developed for the MST sector.

For the editor and reviewers' convenience, we quote the newly added Supplementary Fig. 12-13 about the feature importance for these six developed models as follows:

Supplementary Figure 12. Feature importance (histogram plots) and SHAP summary plots for six application cases of the metal surface treatment sector to predict the total generation quantity of HW. (a)~(f) refer to the application

regions of Shandong Province, Zhejiang Province, Fujian Province, Hebei Province, Jiangsu Province, and Guangdong Province, respectively. In each SHAP summary plot, the features are ranked according to their importance.

Supplementary Figure 13. Importance of input features for six models of application cases in the metal surface treatment sector to predict the total generation quantity of HW. (a) The average relative importance, indicated by the average MAS value of variables in 4 groups of the firm scale, manufacturing processes, the wastewater routine monitoring indicators, and metal emission in wastewater. (b) Ranking of each input feature importance for 6 models. The number inside each cell reflected the ranking of the variable. Variables ranked in the top 5 are marked with *** to denote their crucial importance in model prediction, while variables ranked between 6th and 10th are marked with * to indicate their relatively significant impact.

Based on the comparison of these six regions, we observed consistently high importance attributed to wastewater routine monitoring indicators (wastewater

discharge amount, COD, NH₃-N, P, and N), with the average relative importance of variables in this group ranking 1st for most regions (Supplementary Fig.13). However, significant disparities in feature importance were noted for the groups of manufacturing processes and metal emission in wastewater. For example, while the variable group of metal emission in wastewater (Cr^{VI} and Cr) was most important for the Hebei province, this group of variables were filtered out during feature screening based on casual discovery and feature's importance for the Zhejiang province. This discrepancy was because the enterprises of the MST sector in Hebei province are mainly related to chrome plate, a subsidiary sector of the leather industry which has been listed as one of the pillar industries in this region⁸. Another finding is that a distinctively high importance of manufacturing processes was observed for Guangdong province, owing to its unique industrial layout where electronic waste recycling is well developed. Enterprises engaged in the manufacturing process of circuit board treatment contribute significantly to HW generation in this region⁹. The heterogeneity in feature importance across different regions, even for the same sector, underscores the importance of employing a generic set of variables tailored to the characteristics of sectors in different regions. Subsequently, specific variables used to build models can be screened based on feature engineering, considering the varied importance of features for each application region.

For more detailed information about the spatial heterogeneity regarding feature importance, please refer to lines 504–537 of the revised manuscript.

3. Moreover, beyond the aforementioned model considerations, it may be beneficial to discuss the model's applicability scope and provide specific guidelines for its practical implementation in the final paragraph.

Authors' response:

Suggestion taken. We have added discussions about the model's applicability and guideline for its practical implementation in Section 4.4 'Model applicability, limitation, and future research'.

Regarding the model's applicability scope, our generic framework has been demonstrated being applicable in diverse regions and industrial sectors, provided that model training data is localized, and variables related to

manufacturing processes and water contaminant emission are appropriately screened based on the specific studied sector. For one industrial sector, a generic set of variables tailored to the characteristics of sectors should be employed for different application regions and then specific variables used in each region can be screened based on feature engineering, considering the varied importance of features. In addition, since the model simulates the patterns of HW generation at the firm-level through mining data characteristics, it is suggested to collect as much data as possible to enhance model performance and reliability.

Meanwhile, we divided the specific guidelines for its practical implementation into three major steps, including historic data collection, model development, and model application. Specifically, firstly, the variables involved in the model, referring to the ones related to enterprise characteristics (such as the industrial sector, firm scale, and manufacturing processes) and real-time water contaminant emission, should be tailored based on the characteristics of studied sectors. Regarding the tailored variables, a set of historic data, with a monthly time-resolution, should be collected for subsequent modeling development. Secondly, apply a data-driven methodology incorporating feature engineering, data balancing, and artificial intelligent algorithms to build the model for predicting HW generation quantity at the firm-level for each month. Given the high diversity in HW generation patterns among sectors, it is advisable to develop sector-independent models that can better capture the unique patterns of each sector when the data size for each sector is sufficiently large to ensure confidence in learning. Thirdly, adopt this model to predict the enterprise's HW generation quantity in the next 3 months following the period of collected historic data. Additionally, regular updates of the model every three months are recommended to maintain model performance.

For more detailed information about the discussion on the model's applicability and guideline for its practical implementation, please see texts in lines 638–666 of the revised manuscript.

4. The code cannot be read on the page.

Authors' response:

Thanks for your comments. We have double checked to ensure that the source codes can be accessed freely and read on pages. Please access the GitHub

repository via this link: [<https://github.com/Monchiwjxie/Hazardous-waste-generation.git>].

If there are any further issues with code readability on the webpage, you can download all the data and code. The code is compatible with Python version 3.9.2 and R version 4.3.0. All necessary packages and functions have been clearly indicated within the code. If you require any further assistance, please let me know.

[redacted]

Figure R2. Screenshot of the readable codes on the webpage when log in the GitHub as a guest.

Comments by reviewer #3

Overview:

The authors have solved some of the problems and improved the quality of the manuscript, but the following major problems still need to be addressed before it can be accepted for publication.

Authors' response:

We sincerely appreciate the reviewer's acknowledgment of the enhanced quality of the manuscript following the revision. In response to the remaining major limitation associated with the methodology, we have undertaken further refinement in the revised manuscript. This includes (a) developing three deep learning algorithms (Multilayer Perceptron (MLP), MLP ensemble, and Tabular neural network (TNN)), aiming at advancing our models and (b) modifying the strategy of missing value imputation to prevent information leakage, thereby enhancing the robustness of our approach. Furthermore, we have clarified the requirement to achieve the good generalization across different regions and sectors in the manuscript. We believe these revisions address the concerns raised by the reviewer and contribute to the overall strength of our study. We are grateful for the reviewer's valuable feedback, which has played a pivotal role in improving the quality and rigor of our research.

Below are point-to-point responses in which we explicitly refer to the relevant lines in the revised manuscript/supplementary information to clarify the genuine. The comments appear in black, our responses appear in indent and blue font, and the texts quoted from the revised manuscript/supplementary information appear in italics and green font.

Specific comments:

1. The authors have used five machine learning algorithms to achieve the prediction of HW, how about using currently popular deep learning algorithms? More comparisons and discussions about novel algorithms would make the exposition more complete and strength the importance of this work.

Authors' response:

Suggestion taken. We have incorporated three additional deep learning models for performance comparison, namely Multilayer Perceptron (MLP), MLP

ensemble, and Tabular neural network. In total, eight different models are developed for comprehensive performance evaluation.

MLP consists of fully connected neurons with a nonlinear kind of activation function, organized in at least three layers¹⁰. It is usually trained using the backpropagation method, including two main processes: forward and backward. In the forward process, a neuron’s output is calculated by three steps: the weighting step of multiplying each input feature value by its weight, the sum step of adding them together, and the transfer step of applying an activation function to the sum value¹¹. In the backward propagation, the connection weights between neurons are optimized based on the error in the output compared to the expected result. In our study, Mean Squared Error (MSE) loss and Cross Entropy (CE) loss are utilized to measure the error mentioned above for regressors and classifiers, respectively.

$$\text{MSE Loss} = \frac{1}{n} \sum_{i=1}^n (y_i - \hat{y}_i)^2 \quad (1)$$

$$\text{CE Loss} = \frac{1}{n} \sum_{i=1}^n y_i \cdot \log \hat{y}_i \quad (2)$$

where n is the number of data points in a batch; y_i refer to the observed generation quantity of HW in Eq. (1) and whether HW is generated in Eq. (2), respectively; \hat{y}_i are the predicted generation quantity or the probability of HW generation.

The MLP ensemble method leverages multiple MLPs for ensemble learning. Specifically, we built 5 MLPs trained and used the bagging strategy¹² to aggregate their output. In detail, for the regression model, the predicted value from all the MLP base models were averaged. For the classification model, the dominant classification among the predictions from 5 base MLP models was used as the final result.

Tabular neural network is a special MLP that adopts embedding layers to handle sparse and high-dimensional categorical features^{13,14}. Embedding layer is widely used in NLP and recommendation systems to make up for the critical flaw of one-hot word representation: data-sparse problem and loss of semantic relatedness between words¹⁵. Specifically, the embedding layer is a learnable dense layer that converts one-hot input into low-dimensional dense vectors as

features of the downstream network. Except for the embedding layer, Tabular neural network also uses many other approaches like dropout¹⁶ and Kaiming initialization¹⁷ to improve the performance of the model.

For more detailed information about the algorithms, please see texts in Supplementary Text 8 of the revised Supplementary information.

Performance comparison of the eight models to predict the generation quantity of HW showed that RF model (before data balance: $R^2=0.80$, RMSE=270.35; after data balance: $R^2=0.87$, RMSE=247.40) still achieved the best performance and was thus selected for further model development. Interestingly, three deep learning algorithms did not outperform the RF method in predicting both total generation quantity of HW and MHW. This outcome aligns with findings from a comprehensive study comparing deep learning methods with tree-based models across a standard set of 45 tabular datasets from diverse domains¹⁸. The study demonstrated that tree-based models, such as RF, remained state-of-the-art for medium-sized data (~10K samples), even without considering their superior computational efficiency. This superiority can be attributed to specific characteristics of tabular data, including irregular patterns in the target function and the presence of uninformative features. Neural networks are biased to overly smooth solutions, thereby failing to learn non-smooth and irregular data patterns. In contrast, models based on decision trees, which learn piece-wise constant functions, do not exhibit such a bias¹⁹. In addition, tabular datasets usually contain many uninformative features, but MLP-like architectures are not robust to uninformative features because removing uninformative features will even obviously decrease the model performance²⁰. Even for the deep learning model (TNN before data balance: $R^2=0.81$, RMSE=307.85; Ensemble MLP after data balance: $R^2=0.87$, RMSE=241.09) which performed best in predicting HW generation quantity, their performances were comparable to the RF model, but the time costs of model training (TNN: 5767 seconds; Ensemble MLP: 720 seconds) were much higher than the RF model (12 seconds). Therefore, considering the RF algorithm's ability to achieve accurate predictions with lower computational costs, we have chosen it for subsequent model development.

Table R4. Performance comparison of the combined models developed from eight algorithms to predict the generation quantity of HW and MHW after data

balance.

		Models	R²	RMSE
Predicting the total generation quantity of HW	Machine Learning	GBDT	0.81	298.05
		XGB	0.81	296.17
		SVM	0.81	295.93
		KNN	0.80	290.70
		RF	0.87	247.40
Deep Learning	MLP	0.86	249.99	
	Ensemble MLP	0.87	241.09	
	TNN	0.84	295.82	
Predicting the generation quantity of MHW	Machine Learning	GBDT	0.48	116.34
		XGB	0.50	118.12
		SVM	0.54	87.35
		KNN	0.45	106.43
		RF	0.85	47.50
Deep Learning	MLP	0.67	68.63	
	Ensemble MLP	0.21	108.09	
		TNN	0.56	96.32

Note that the test dataset used for evaluating the performance of all models is the same. Data balance is conducted only on the training dataset.

For more detailed information about the model comparison and discussion, please see Supplementary Table 10–12 of the revised supplementary information and texts in lines 203–230 of the revised manuscript.

2. As the authors show the results in lines 452–454, the model's R2 is 0.69 and 0.72 in the cases of Shandong Province and Zhejiang Province, respectively, while the value is 0.87 in the case of Jiangsu Province. What are the reasons for the differences in the model's effectiveness in different cases? How to solve this problem of poor generalization?

Authors' response:

Thanks for your comments. It is true that the performance of models developed for Shandong Province and Zhejiang Province are relatively lower than that of Jiangsu case previously developed from monthly observations. This might be primarily attributed to the smaller data size (the Shandong case: 190 observations; the Zhejiang case: 396 observations) due to the data availability.

If possible, collecting more data has the potential to improve the performance of data-driven approaches because the expansion of the dataset compensates for the loose structure of the current small dataset and the omission of potential information, thereby improving the representativeness of the data and inducing the underlying model to better characterize the actual distribution²¹. For more detailed information about the discussion on the differences in the model's performance among various cases, please see texts in lines 483–493 of the revised manuscript.

Furthermore, in Section 4.4 where we discussed the model's applicability, we emphasized the key requirements to achieve robust generalization across different regions and sectors. These requirements primarily involve: (a) utilizing localized data for model training, (b) appropriately screening variables related to manufacturing processes and water contaminant emission based on the specific sector under study, and (c) collecting as much data as possible to improve the model performance. For more detailed information about the discussion on the model applicability and requirements to achieve the good generalization across different regions and sectors, please see texts in lines 639–649 of the revised manuscript.

3. As shown in Table Supplementary Table 21, some of the indicators are missing more than 80%, does the methods of missing value imputation cause information leakage, in other words, introducing information from the test dataset into the training dataset? Please kindly know that information leakage is one of the major reasons that cause the fictitious good prediction performance (Sustainable Cities and Society 94 (2023) 104541). The above related paper and other closely related ones are suggested to cited and discussed.

Authors' response:

Suggestion taken. We agreed with the reviewer that the information leakage might exist during current missing value imputation strategy because the process of missing value imputation was made before data splitting. It might cause the risk of unintentionally exposing sensitive information of testing data to the training data model. To overcome this limitation, we have modified the procedure of data processing to split the data into training and testing sets before performing any preprocessing steps. In this way, there's no chance of

information from the testing set leaking into the training set during model development.

Specifically, the whole dataset was randomly split into a training dataset and a testing dataset before any preprocessing steps were undertaken. The training dataset was considered as known data, and the missing values in both the training and testing dataset were imputed based on the information from the known data rather than the whole raw dataset²².

Regarding the training dataset, missing values of the variables of wastewater routine monitoring indicators (COD, NH₃-N, N, and P) were imputed by multiplying the contaminant emission intensity by the wastewater discharge amount. In detail, the contaminant emission intensity, referring to the contaminant emission per wastewater flow, was the median value of each firm. If the enterprise had no records of this indicator, the median value at the industrial sectors level was used. Similarly, the missing value for wastewater pH was imputed by the median value of the enterprise or the industrial sector.

In terms of the testing dataset, the missing value imputation strategy is similar to the training dataset, but the median value of contaminant emission intensity and wastewater pH was calculated based on the training dataset.

The detailed information about missing value imputation could be found in Supplementary Text 1 in the revised supplementary information.

Regarding that some variables (i.e., N and P) exhibited high percentages of missing values, even exceeding 80%, the principal logic of imputing these missing values through multiplying the contaminant emission intensity by the wastewater discharge amount might cause tight correlation between these variables and the feature of wastewater discharge amount. If all these variables with strongly tight correlations were used as the input features, it is potential to induce features redundancy. Therefore, to mitigate this risk, we employed feature engineering techniques to select input features while considering correlations between variables. Specifically, among variable clusters exhibiting high correlations (Spearman's correlation > 0.6), we retained only the variables with the highest importance ranking to prevent redundancy. This approach ensured that variables such as wastewater discharge amount, N, and P were not

simultaneously used as input features, thereby avoiding redundancy. The finalized input features for the combined and sector-independent models are presented in Table R5.

Table R5. Input features of the combined and sector-independent models.

Models		Input features	
Models to predict the total generation quantity of HW	The Combined model	process_1, process_2, process_4, process_6, process_8, process_9, process_10, process_11, industrial sector, firm scale, pH, wastewater discharge amount, COD, Fe, Ni	
	OCM	process_5, process_6, process_7, process_8, process_9, process_10, process_11, firm scale, pH, wastewater discharge amount	
	CPM	process_5, process_6, process_7, process_8, process_9, process_10, process_11, firm scale, N	
	SCP	process_2, process_3, process_5, process_6, process_7, process_8, process_9, process_10, process_11, firm scale, pH, P	
	Sector independent models	SRP	process_1 process_2, process_3, process_9, process_10, process_11, firm scale, pH, N
		MWR	process_1 process_2, process_3, process_9, process_10, process_11, firm scale, pH, P
		MST	process_1 process_2, process_3, process_9, process_10, process_11, firm scale, pH, NH ₃ -N, Ni
		ECM	process_2, process_3, process_4, process_6, process_8, process_9, process_10, process_11, firm scale, pH, wastewater discharge amount, Ni
		BEG	process_2, process_3, process_6, process_7, process_9, process_10, process_11, firm scale, pH, COD
		EGU	process_2, process_3, process_4, process_6, process_8, process_9, process_10, process_11, firm scale, pH, P, NH ₃ -N
Models to predict the generation quantity of MHW	The Combined model	process_1, process_2, process_4, process_6, process_8, process_9, process_10, process_11, industrial sector, firm scale, pH, COD, Fe, Ni	
	SRP	process_1, process_2, process_3, process_9, process_10, process_11, firm scale, pH, NH ₃ -N, Ni, Cr	
	MWR	process_1, process_2, process_3, process_4, process_9, process_10, firm scale, pH, P, Ni, Zn	
	MST	process_1, process_2, process_3, process_9, process_10, process_11, firm scale, pH, water, Ni, Cu, Fe, Cr, Cr6	

Models	Input features
ECM	process_1, process_2, process_3, process_4, process_6, process_9, process_10, process_11, firm scale, pH, COD, NH ₃ -N, Ni, Cu

The detailed information about input features after feature engineering for the combined and sector-independent models could be found in Supplementary Table 23 and 24 in the revised supplementary information.

4. The model framework shown in Figure 2 is incomplete, for example, the Data Preprocessing also includes missing value imputation. The completeness of the model framework is crucial for readers to understand and use the authors' ensemble model.

Authors' response:

Suggestion taken. We have revised the model framework in Figure 2 to make it more complete. The major modification existed in the part of data collection and preprocessing, where we precisely showed the procedure of collected variables, data quality control, missing value imputation, and outlier rejection.

For the editor and reviewers' convenience, we quote the newly revised Figure 2 as follows:

[redacted]

Figure 2. The model framework for predicting the total generation quantity of HW and one specific category. (a) Data of variables which could indicate static characteristics and real-time activities of enterprises are collected and preprocessed. (b) Feature engineering which incorporated the causal discovery, importance, and correlation analysis was conducted to screen input features. (c) A regression model is developed directly from the training dataset with data balance to predict the total generation quantity of HW. (d) An ensemble model coupling the classification and regression model is developed to predict the generation quantity of one category of HW. The binary classification model could determine whether the generation quantity of this category is 0. The regression model is developed to predict the specific value when the generation quantity of this category was > 0 based on the classification model results. (e) After model development, performance validation is conducted on the test dataset. Additionally, feature exploration is performed on the trained models to investigate the impact of input features on the target output.

References

- 1 Zhong, S. *et al.* Machine Learning: new ideas and tools in environmental science and engineering. *Environmental Science & Technology* **55**, 12741-12754 (2021). <https://doi.org/10.1021/acs.est.1c01339>
- 2 Xu, P. *et al.* Fertilizer management for global ammonia emission reduction. *Nature* **626**, 792-798 (2024). <https://doi.org/10.1038/s41586-024-07020-z>
- 3 Guerra, C.A. *et al.* Global hotspots for soil nature conservation. *Nature* **610**, 693-698 (2022). <https://doi.org/10.1038/s41586-022-05292-x>
- 4 Hsu, A. *et al.* Predicting European cities' climate mitigation performance using machine learning. *Nature Communications* **13**, 7487 (2022). <https://doi.org/10.1038/s41467-022-35108-5>
- 5 Barnett, J.W. *et al.* Designing exceptional gas-separation polymer membranes using machine learning. *Science Advances* **6**, eaaz4301 (2020). <https://doi.org/doi:10.1126/sciadv.aaz4301>
- 6 Yu, F. *et al.* Deep exploration of random forest model boosts the interpretability of machine learning studies of complicated immune responses and lung burden of nanoparticles. *Science Advances* **7**, eabf4130 (2021). <https://doi.org/doi:10.1126/sciadv.abf4130>
- 7 Wang, Y. *et al.* Persistent and enhanced carbon sequestration capacity of alpine grasslands on Earth's Third Pole. *Science Advances* **9**, eade6875 (2023). <https://doi.org/doi:10.1126/sciadv.ade6875>
- 8 Li, X. *et al.* Dynamic Optimized Cleaner Production Strategies to Improve Water Environment and Economic Development in Leather Industrial Parks: A Case Study in Xinji, China. *Sustainability* **11**, 6828 (2019).
- 9 Yuan, Q. *et al.* Synergistic utilization mechanism of e-waste in regions with different levels of development: A case study of Guangdong Province. *Journal of Cleaner Production* **380**, 134855 (2022). <https://doi.org/https://doi.org/10.1016/j.jclepro.2022.134855>
- 10 Cybenko, G. Approximation by superpositions of a sigmoidal function. *Mathematics of Control, Signals and Systems* **2**, 303-314 (1989). <https://doi.org/10.1007/BF02551274>
- 11 Taud, H. & Mas, J.F. Multilayer Perceptron (MLP). *Geomatic Approaches for Modeling Land Change Scenarios*, 451-455 (2018). https://doi.org/10.1007/978-3-319-60801-3_27
- 12 Efron, B. Bootstrap Methods: Another Look at the Jackknife. *Breakthroughs in Statistics: Methodology and Distribution*, 569-593 (1992). https://doi.org/10.1007/978-1-4612-4380-9_41
- 13 Cheng, H.T. *et al.* Wide & Deep Learning for Recommender Systems. *Proceedings of the 1st Workshop on Deep Learning for Recommender Systems* (2016).
- 14 Guo, H. *et al.* DeepFM: A Factorization-Machine based Neural Network for CTR Prediction. *ArXiv abs/1703.04247* (2017).
- 15 Wang, S., Zhou, W. & Jiang, C. A survey of word embeddings based on

- deep learning. *Computing* **102**, 717-740 (2020).
<https://doi.org/10.1007/s00607-019-00768-7>
- 16 Srivastava, N. *et al.* Dropout: a simple way to prevent neural networks from overfitting. *The Journal of Machine Learning Research* **15**, 1929-1958 (2014).
- 17 He, K. *et al.* Delving Deep into Rectifiers: Surpassing Human-Level Performance on ImageNet Classification. *Proceedings of the 2015 IEEE International Conference on Computer Vision (ICCV)*, 1026–1034 (2015).
<https://doi.org/10.1109/iccv.2015.123>
- 18 Grinsztajn, L., Oyallon, E. & Varoquaux, G. Why do tree-based models still outperform deep learning on typical tabular data? *Proceedings of the 36th International Conference on Neural Information Processing Systems*, Article 37 (2024).
- 19 Agarwal, R. *et al.* Neural Additive Models: Interpretable Machine Learning with Neural Nets. *ArXiv* **abs/2004.13912** (2020).
- 20 Andrew Y, N. Feature selection, L1 vs. L2 regularization, and rotational invariance. *Proceedings of the twenty-first international conference on Machine learning* (2004).
- 21 Lin, L. *et al.* An attribute extending method to improve learning performance for small datasets. *Neurocomputing* **286**, 75-87 (2018).
<https://doi.org/https://doi.org/10.1016/j.neucom.2018.01.071>
- 22 Yang, M. *et al.* Predicting extraction selectivity of acetic acid in pervaporation by machine learning models with data leakage management. *Environmental Science & Technology* **57**, 5934-5946 (2023). <https://doi.org/10.1021/acs.est.2c06382>

REVIEWERS' COMMENTS

Reviewer #2 (Remarks to the Author):

The methodology for estimating hazardous waste from wastewater data outlined in this paper holds significant methodological importance, and its demonstration process has been refined to perfection. The findings have potential for publication in Nature Communications. The accompanying charts and data are comprehensive, and the code works well. Nonetheless, the descriptive section of the code is written in Chinese, raising uncertainty about its conformity to NC standards. Overall, we highly recommend accepting the paper.

Reviewer #2 (Remarks on code availability):

The code and data are working fine.

Reviewer #3 (Remarks to the Author):

The authors have addressed all the issues raised by me, and I have no further question.

Itemized Replies to Review Comments

(Manuscript Number: NCOMMS-23-45555B)

Comments by reviewer #2

Overview:

The methodology for estimating hazardous waste from wastewater data outlined in this paper holds significant methodological importance, and its demonstration process has been refined to perfection. The findings have potential for publication in Nature Communications. The accompanying charts and data are comprehensive, and the codeworks well. Nonetheless, the descriptive section of the code is written in Chinese, raising uncertainty about its conformity to NC standards. Overall, we highly recommend accepting the paper.

Authors' response:

We would like to thank the reviewer for the positive feedback on the methodology importance, demonstration processes, and comprehensive results.

Regarding the concern about the descriptive section of the code being written in Chinese, we have translated all Chinese code comments into English to ensure compliance with the journal's standards. The revised code has also been uploaded in the public repository.

Specific comments:

1. The code and data are working fine.

Authors' response:

We would like to thank the reviewer for taking the time to review our codeworks.

Comments by reviewer #3

Overview:

The authors have addressed all the issues raised by me, and I have no further question.

Authors' response:

We sincerely thank the reviewer for the effort and comments to improve the quality of this manuscript.